# Identification of a subset of immunosuppressive P2RX1-negative neutrophils in pancreatic cancer liver metastasis

Xu Wang[1,2,4], Li-Peng Hu[1,4], Wei-Ting Qin[1,4], Qin Yang[1,4], De-Yu Chen[2,4], Qing Li[1], Kai-Xia Zhou[1], Pei-Qi Huang[1], Chun-Jie Xu[1], Jun Li[1], Lin-Li Yao[1], Ya-Hui Wang[1], Guang-Ang Tian[1], Jian-Yu Yang[3], Min-Wei Yang[3], De-Jun Liu[3], Yong-Wei Sun [3✉], Shu-Heng Jiang [1✉], Xue-Li Zhang [1✉] & Zhi-Gang Zhang [1✉]

The immunosuppressive microenvironment that is shaped by hepatic metastatic pancreatic ductal adenocarcinoma (PDAC) is essential for tumor cell evasion of immune destruction. Neutrophils are important components of the metastatic tumor microenvironment and exhibit heterogeneity. However, the specific phenotypes, functions and regulatory mechanisms of neutrophils in PDAC liver metastases remain unknown. Here, we show that a subset of P2RX1-negative neutrophils accumulate in clinical and murine PDAC liver metastases. RNA sequencing of murine PDAC liver metastasis-infiltrated neutrophils show that P2RX1-deficient neutrophils express increased levels of immunosuppressive molecules, including PD-L1, and have enhanced mitochondrial metabolism. Mechanistically, the transcription factor Nrf2 is upregulated in P2RX1-deficient neutrophils and associated with PD-L1 expression and metabolic reprogramming. An anti-PD-1 neutralizing antibody is sufficient to compromise the immunosuppressive effects of P2RX1-deficient neutrophils on OVA-activated OT1 CD8+ T cells. Therefore, our study uncovers a mechanism by which metastatic PDAC tumors evade antitumor immunity by accumulating a subset of immunosuppressive P2RX1-negative neutrophils.

[1] State Key Laboratory of Oncogenes and Related Genes, Shanghai Cancer Institute, Ren Ji Hospital, School of Medicine, Shanghai Jiao Tong University, Shanghai, P.R. China. [2] Department of Radiation Oncology, Institute of Oncology, Affiliated Hospital of Jiangsu University, Zhenjiang, P.R. China. [3] Department of Biliary-Pancreatic Surgery, Ren Ji Hospital, School of Medicine, Shanghai Jiao Tong University, Shanghai, P.R. China. [4] These authors contributed equally: Xu Wang, Li-Peng Hu, Wei-Ting Qin, Qin Yang, De-Yu Chen. ✉email: syw0616@126.com; shjiang@shsci.org; xlzhang@shsci.org; zzhang@shsci.org

Pancreatic ductal adenocarcinoma (PDAC) is the fourth most common cause of death from cancer, with a 5-year survival rate of 8%[1]. PDAC remains one of the most lethal cancers in part due to its aggressive metastatic nature[2]. The liver is the most common site of distant metastases of PDAC, and more than half of PDAC patients are diagnosed with liver metastases. Whole-genome sequencing studies revealed limited genetic heterogeneity between primary and metastatic tumors, with remarkably uniform mutations in PDAC driver genes, including *KRAS*, *TP53*, and *SMAD4*[3]. However, when comparing the transcriptome data of primary and metastatic tumors that contain microenvironment tissues, major differences were uncovered[4]. To establish metastatic tumors in liver tissues, the cells must successfully negotiate complex steps within a foreign microenvironment. The acquisition of genomic alterations does not fully explain the process of liver metastases, which is likely to be influenced not only by tumor cells but also by the tumor microenvironment.

PDAC is nonimmunogenic and characterized by a desmoplastic tumor microenvironment, with high numbers of fibroblasts and extracellular matrix deposition[5]. Although liver metastases were also identified with a similar desmoplastic stroma, recent studies observed that the metastatic tumor microenvironment possesses a relatively abundant infiltration of immune cells[6]. Therefore, shaping an immunosuppressive microenvironment is a critical step by which metastatic tumors evade antitumor immunity and colonize foreign tissues. The immunosuppressive microenvironment at the primary tumor site has been studied for decades; however, relatively little is known about how the hepatic local immune microenvironment is mediated by metastatic tumors.

Neutrophils have long been recognized as a homogeneous cell population that constitutes the first line of defense against invading pathogens. Recently, accumulating evidence has revealed the essential role of neutrophils in the tumor microenvironment, with both tumor-suppressing and tumor-promoting effects[7,8]. However, the specific phenotypes and mechanisms that characterize this heterogeneity remain unclarified.

Extracellular nucleotides (particularly ATP) and adenosine (ADO) are major biochemical constituents of tumor microenvironment[9]. By activating tumoral purinergic P2- and P1-receptors, extracellular purines facilitate the tumor growth directly[10]. In addition, host immune cells express abundant P2- and P1- receptors, and extracellular purines are essential for mediating immune cells trafficking and immune phenotypes[11]. The purinergic receptor P2RX1 is an ATP-gated ion channel and participates in smooth muscle contraction and immunity[12,13].

In this work, we identify a subset of P2RX1-negative (P2RX1−) neutrophils that is mobilized and recruited in PDAC liver metastases. P2RX1 deficiency shapes the immunosuppressive nature of this subset of neutrophils and promotes PDAC liver metastases. Further studies demonstrate that upregulated activity of the anti-inflammatory transcription factor Nrf2 contributes to the generation of the immunosuppressive phenotype of P2RX1-neutrophils.

## Results

**Reduced P2RX1 is associated with an immunosuppressive microenvironment in PDAC liver metastases**. To investigate specific transcriptome alterations that might influence PDAC liver metastases and its microenvironment, bioinformatics analyses were performed using a Gene Expression Omnibus (GEO) dataset (GSE71729) containing bulk-tissue microarray transcriptome data from primary PDAC, adjacent pancreas, liver metastatic PDAC and adjacent liver tissues (Supplementary

Fig. 1a). Differential analysis comparing primary PDAC and adjacent pancreas identified a set of differentially expressed genes, 82 of which were involved in immune responses (Fig. 1a). In comparison, 320 differentially expressed genes between liver metastases and adjacent liver tissues were implicated in immune responses (Fig. 1a). We also found that primary PDAC samples and liver metastatic tissues showed significant differences in immune responses.

To identify specific immune cells linked to PDAC liver metastasis, the immunome, a compendium of 28 immune cell types that infiltrated in tumors, was analyzed[14]. The results showed that immune cell infiltration was intensively reprogrammed in primary and metastatic PDAC (Fig. 1b). We noticed that antitumor immune cells, including activated CD8+, central memory CD8+, effector memory CD8+ and T helper type 1 (Th1) cells, were markedly downregulated in PDAC liver metastases compared to those in primary PDAC and adjacent liver tissues, whereas the classically recognized protumoral Th2 cells were upregulated (Fig. 1b). Additionally, gene ontology (GO) analysis revealed substantial enrichment of certain antitumor immunity-associated pathways, including antigen processing and presentation and the T cell receptor signaling pathway, that were downregulated compared to those of adjacent nontumoral liver tissue, whereas tumor growth-related pathways, such as DNA replication and epithelial cell proliferation, were upregulated (Fig. 1c). These analyses demonstrated a local immunosuppressive microenvironment in hepatic metastatic PDAC, which might facilitate local tumor growth. Emerging evidence suggests that neutrophils play important roles in tumor metastasis[15]. We found that neutrophils were rare in primary PDAC and adjacent pancreas (Fig. 1b). However, in PDAC liver metastases and adjacent liver tissues, neutrophils were among the most abundant immunocytes, which suggested that they might have potential roles in PDAC liver metastasis. Apart from Th1 and Th2 cells, the role of Th17 cells in tumor microenvironment has recently attracted much attention, with both promotive[16] and suppressive[17] effects on tumor metastasis having been reported. Our analyses showed that primary PDAC and PDAC liver metastases had more Th17 cells as compared to adjacent pancreas and adjacent liver tissues, respectively (Fig. 1b). The specific function of Th17 in PDAC progression remains to be explored.

Interestingly, we noted that the purinergic receptor signaling pathway and response to ATP-related gene expression were dysregulated in PDAC liver metastases (Fig. 1b). Given the essential role of purinergic signaling in manipulating the immune response and tumor growth[11,18], we next assessed the relevance of purinergic signaling-associated molecules in PDAC liver metastases, which included 18 purinergic receptors, 46 purine transmitters and 37 purine hydrolases. The results showed that these molecules were thoroughly reprogrammed, with 26 significantly upregulated and 46 significantly downregulated molecules (Supplementary Fig. 1b, c). Venn diagram analysis indicated that *ADORA2B*, a well-recognized immunosuppressive purinergic receptor, overlapped in upregulated gene lists, whereas *P2RX1* overlapped in the downregulated gene lists when comparing liver metastases *vs* adjacent liver tissue, primary PDAC *vs* liver metastases and primary PDAC *vs* adjacent pancreas (Fig. 1d, Supplementary Fig. 1b and c).

Next, correlations between PDAC metastatic immune components and purinergic signaling molecules were analyzed (Supplementary Fig. 2a). We observed that *ADORA2B* was negatively correlated with antitumor immunocyte infiltration, whereas *P2RX1* was positively correlated with antitumor immunocyte infiltration (Supplementary Fig. 2a). The roles of *ADORA2B* in promoting tumor metastasis have been well characterized[19,20]; however, the roles of *P2RX1* in PDAC progression or metastasis

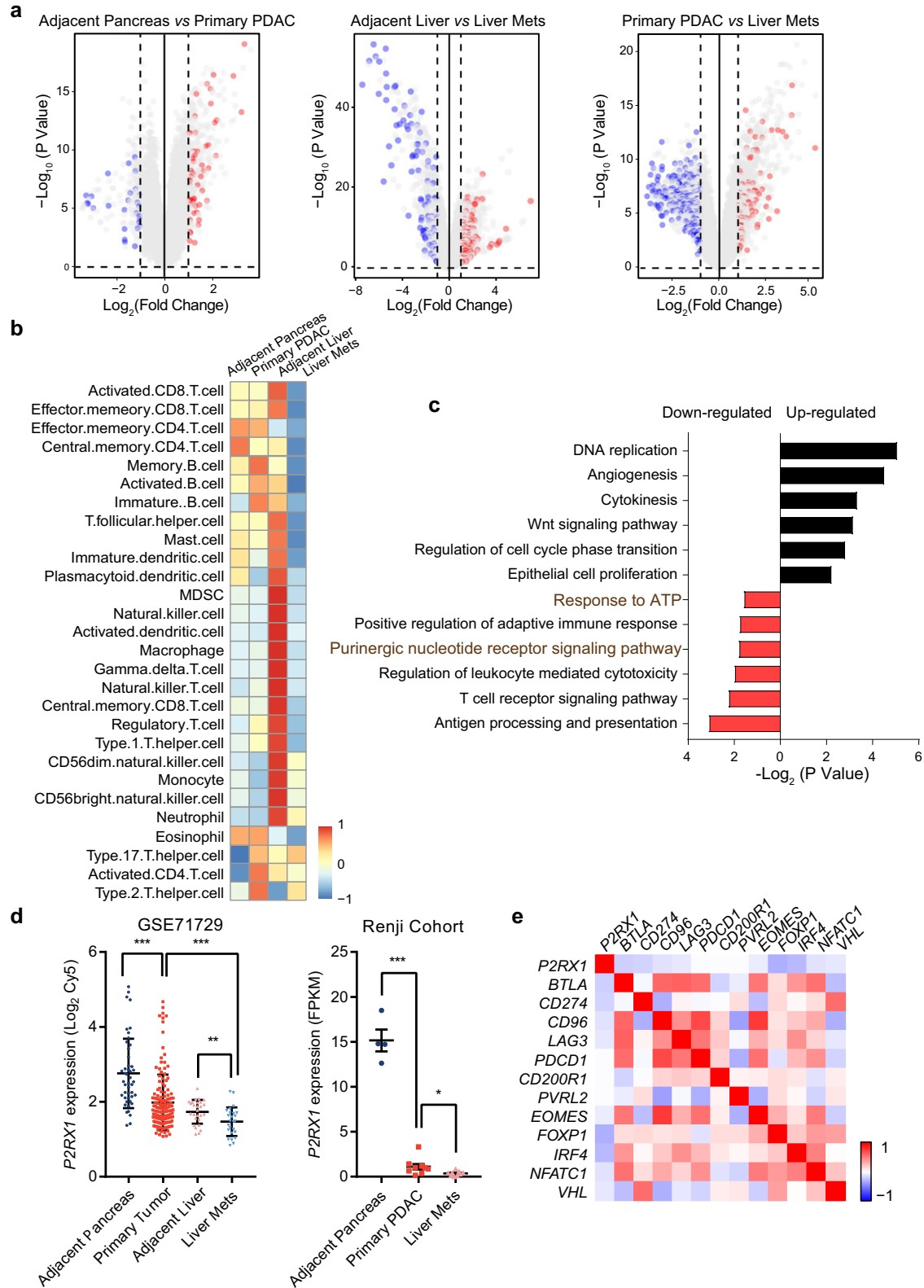

remain unknown. Gene set enrichment analysis (GSEA) of metastatic PDAC indicated that *P2RX1* positively correlated with purinergic ATP-associated signaling pathways, TNF superfamily production, IFN-γ secretion and myeloid cell activation (Supplementary Fig. 2b). Moreover, *P2RX1* was negatively correlated with several inhibitory immune checkpoint molecules, including

*CD274* (*PD-L1*), *LAG3* and *BTLA* (Fig. 1e). The reduced expression of *P2RX1* in PDAC liver metastases was further confirmed by our Renji Cohort RNA-seq data (GSE151580) (Fig. 1d). These results indicated that reduced *P2RX1* in PDAC liver metastases contributes to the progression of PDAC metastases by inducing immunosuppression.

**Fig. 1 Reduced P2RX1 is associated with an immunosuppressive microenvironment in clinical PDAC liver metastases. a** Volcano plots of differential gene expression in 145 primary PDAC, 46 adjacent pancreases, 25 liver metastases and 27 adjacent livers. Red dots represent upregulated immune-related genes, and blue dots represent downregulated immune-related genes. **b** Immunome analyses of 26 infiltrating immune cell types in adjacent pancreas, primary PDAC, adjacent liver tissue and metastatic PDAC samples. **c** GO Biological Process analyses of differentially expressed genes between adjacent liver tissue and metastatic PDAC samples. **d** Expression analyses of P2RX1 in the adjacent pancreas, primary PDAC, adjacent liver tissue and metastatic PDAC samples from the GSE71729 and Renji cohorts. **e** Correlation analyses between P2RX1 and immune checkpoint molecules in metastatic PDAC samples. Bars represent mean ± standard deviation in (**d**). *$P < 0.05$, **$P < 0.01$, and ***$P < 0.001$, by one-way ANOVA and Tukey's multiple comparisons test (**d** left), or Student's $t$ test (**d** right). Source data are provided as a Source data file.

**P2RX1 deficiency promotes PDAC liver metastases and upregulates immune exhaustion-related gene expression**. The public single-cell RNA-seq database revealed that *P2RX1* was barely detected in hepatic cells of liver tissue, or pancreatic ductal cells of pancreas tissue (Supplementary Fig. 3a). The Human Protein Atlas (HPA) tissue RNA-seq database showed that *P2RX1* was mainly distributed in smooth muscle tissues and immune organs (Supplementary Fig. 3b). The HPA cell line RNA-seq database that contains 64 types of cell lines showed that *P2RX1* was only detected in immune cell lines but not in PDAC or other types of cell lines (Supplementary Fig. 3c). We also confirmed that P2RX1 protein was not detected in PDAC cell lines (Supplementary Fig. 3d), suggesting that PDAC liver metastasis might be irrelevant to tumor-derived P2RX1. To further address the issue of the contribution of host-derived P2RX1 to PDAC liver metastasis, we generated P2RX1 knockout ($P2rx1^{-/-}$) mice with the CRISPR/Cas9 system (Fig. 2a, b). The $Kras^{G12D/+}/Trp53^{R172H/+}/Pdx-1$-Cre (KPC) mouse-derived syngeneic PDAC cell line, referred to as KPC, was intrasplenically injected to seed these livers of WT and $P2rx1^{-/-}$ mice. The fate of the tumor cells was followed by in vivo bioluminescence imaging at sequential time points. We found that liver metastatic tumors grew significantly faster in $P2rx1^{-/-}$ mice than in WT mice (Fig. 2c–e). The metastatic liver weight of $P2rx1^{-/-}$ mice was heavier than that of WT mice (Fig. 2f). In addition, worse survival was observed in $P2rx1^{-/-}$ mice (Fig. 2g). Similar results were obtained when a highly selective P2RX1 antagonist, NF449, was administrated in WT mice (Supplementary Fig. 4a). These results demonstrated that blockage of P2RX1 contributed to PDAC liver metastasis.

To investigate the underlying mechanisms of P2RX1 deficiency on liver metastasis, we performed comparative transcriptomics analysis of normal liver (day 0) and two sequential stages (days 3 and 17) of liver metastases. Differential analysis showed that 270 genes were upregulated and 234 genes were downregulated at day 3 in P2RX1-deficient mice (Supplementary Fig. 4b). In addition, differentially expressed genes were markedly increased at day 17, with 1627 genes upregulated and 1481 genes downregulated (Supplementary Fig. 4b). Principal component analysis (PCA) of RNA-seq data revealed global transcriptome changes at day 17 in PDAC liver metastasis of WT and $P2rx1^{-/-}$ mice (Fig. 2h). GO analysis revealed that the tumor growth and immunosuppression pathways were upregulated in $P2rx1^{-/-}$ mice (Supplementary Fig. 4c). Ki-67 immunohistochemical (IHC) staining confirmed the vigorous growth of metastatic tumors in $P2rx1^{-/-}$ mice (Fig. 2i and Supplementary Fig. 4d). Flow cytometry revealed that the composition of infiltrated immune cells changed thoroughly between liver metastasis and adjacent liver tissue (Supplementary Fig. 4e). Given that P2RX1 was negatively correlated with immune exhaustion-associated gene markers in clinical PDAC metastases (Fig. 1b), we inspected immune exhaustion-related genes in WT and $P2rx1^{-/-}$ mice. The results showed that the expression of exhaustion-related genes, including *Pdcd1* (PD-1) and *CD274* (PD-L1), was significantly upregulated in liver metastases of WT mice and was further increased in $P2rx1^{-/-}$ mice (Fig. 2j). Taken together, the results from our in vivo PDAC

liver metastatic model confirmed that P2RX1 deficiency promoted tumor growth and immune exhaustion in the liver.

**P2RX1-negative neutrophils accumulate in the microenvironment of PDAC liver metastases and promote metastatic tumor growth**. Because P2RX1 is predominantly distributed in immune organs and P2RX1 is related to antitumor responses, we hypothesized that immunocyte-derived P2RX1 played a role in PDAC liver metastasis. No significant differences of immunocytes distribution were observed between naïve WT and $P2rx1^{-/-}$ mice (Supplementary Fig. 5a). Then, flow cytometry was performed to inspect the expression of P2RX1 in mouse spleen immunocytes. The results showed that Ly6G+ neutrophils had the highest expression intensity and frequency (Fig. 3a and Supplementary Fig. 5b). Next, P2RX1 expression in metastatic liver tissues was examined. We found that P2RX1+ cells were predominantly coexpressed with CD45+ immune cells in the adjacent liver (Fig. 3b and Supplementary Fig. 5c). However, P2RX1+ CD45+ immune cells were significantly reduced in the liver metastases of PDAC. Immune cell subtype markers further showed that P2RX1 expression on neutrophils was significantly decreased in liver metastases compared to adjacent livers (Fig. 3c and Supplementary Fig. 5d, e). To confirm the flow cytometry results, we performed immunofluorescence staining of P2RX1 and neutrophil markers in clinical PDAC liver metastases, KPC mouse liver metastases, and KPC cell intrasplenic injection-induced liver metastases. The results showed that P2RX1 predominantly colocalized with neutrophil markers in adjacent liver tissues but was absent from neutrophil markers in metastatic tumor tissues (Fig. 3d–f and Supplementary Fig. 5f). These results suggested that P2RX1− neutrophils accumulated in PDAC liver metastases.

To determine the functional significance of neutrophils in metastasis, we depleted neutrophils with anti-Ly6G antibodies in WT and $P2rx1^{-/-}$ mice 1 day before the intrasplenic injection of KPC cells. The results showed that the formation of liver metastases in WT mice was reduced by anti-Ly6G antibodies (Supplementary Fig. 5g). In addition, tumor growth in the livers of $P2rx1^{-/-}$ mice was remarkably impaired to a comparable level as that in WT mice, suggesting that the promotional effects of P2RX1 deficiency on liver metastases was related to neutrophils. When we depleted neutrophils 4 days after the intrasplenic injection of KPC cells, only minimal reduction of liver metastases was observed in WT mice (Supplementary Fig. 5h). Neutrophils-involved tumor seeding in the early metastasis cascade may be the potential causes[21,22]. Interestingly, a significant reduction of metastatic burden was still obtained in $P2rx1^{-/-}$ mice and no differences of metastatic tumor growth were observed between WT and $P2rx1^{-/-}$ mice receiving anti-Ly6G antibodies at 4 days (Supplementary Fig. 5h), indicating that P2RX1-involved PDAC liver metastasis is neutrophils related. Next, four groups of bone marrow chimeras were generated using WT mice and $P2rx1^{-/-}$ mice: WT → WT, $P2rx1^{-/-}$ → $P2rx1^{-/-}$, WT → $P2rx1^{-/-}$, and $P2rx1^{-/-}$ → WT. We found that tumors established in mice reconstituted with $P2rx1^{-/-}$ bone marrow cells grew significantly faster than those of mice reconstituted with WT bone

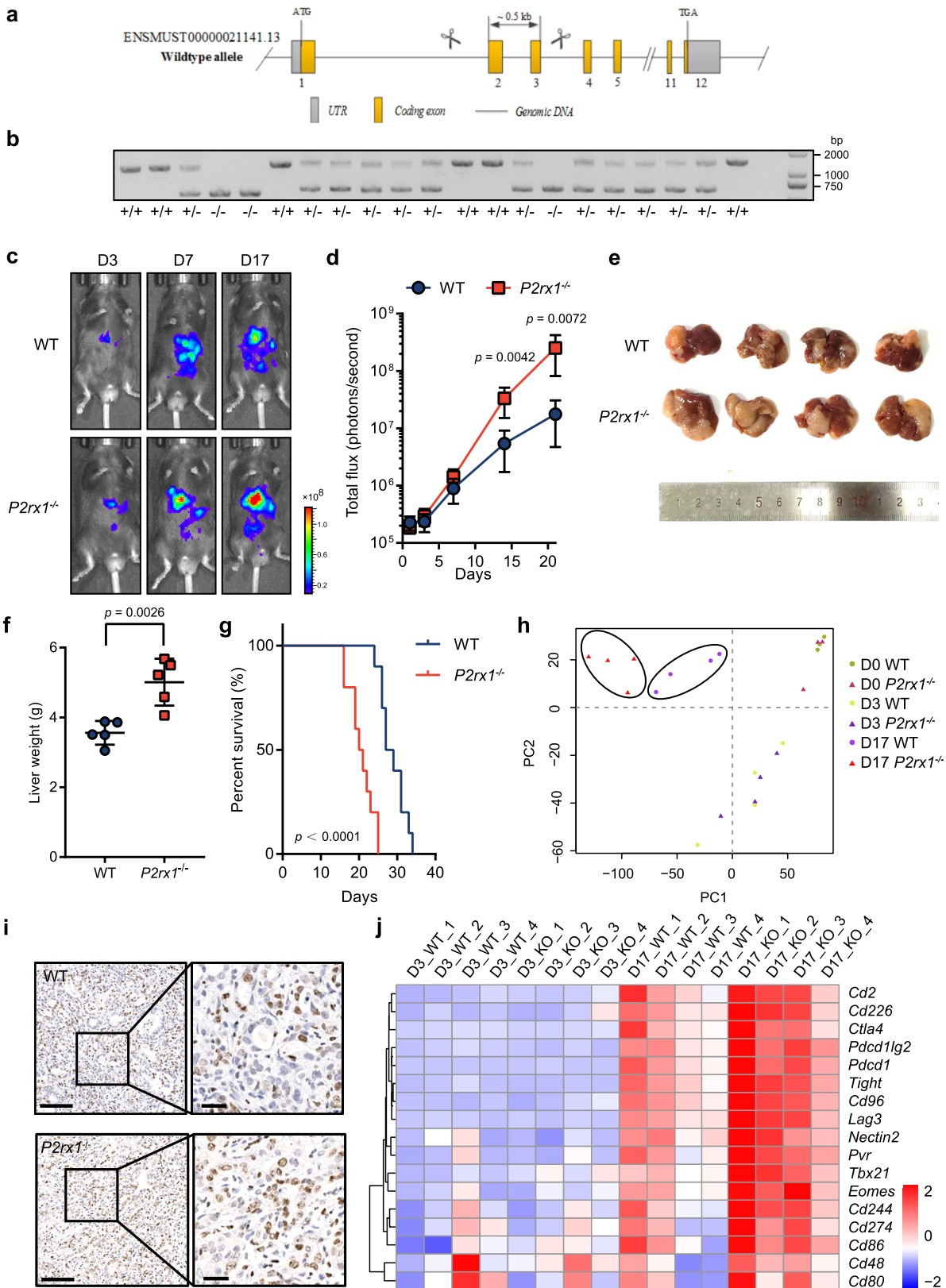

marrow cells (Fig. 3g, h). Then, neutrophils were depleted by anti-Ly6G antibodies in the WT → $P2rx1^{-/-}$ and $P2rx1^{-/-}$ → WT groups. We found that compared to the level of metastases in the $P2rx1^{-/-}$ → WT group, neutrophil depletion resulted in a reduction in liver metastases to comparable levels as tumor growth in the WT → $P2rx1^{-/-}$ group (Fig. 3g, h). These results

indicated that the accumulation of P2RX1-deficient neutrophils was required to promote PDAC liver metastasis.

**PDAC liver metastasis systematically mobilizes and recruits P2RX1-negative neutrophils.** To better understand the systemic

**Fig. 2 P2RX1 deficiency promotes PDAC liver metastases and upregulates immune exhaustion-related gene expression. a**, **b** $P2rx1^{-/-}$ mice were generated using CRISPR/Cas9 system. Schematic diagram was shown in (**a**) and genotyping results were shown in (**b**) (representative result from three independent experiments). **c**, **d** KPC cells were intrasplenically injected to seed livers of WT and $P2rx1^{-/-}$ mice, and in vivo imaging was performed at sequential times. Representative images are shown in (**c**), and quantitative results are shown in (**d**) ($n = 6$ per group, three independent experiments). **e** Representative images of liver metastatic samples harvested at day 17. **f** Liver weight of liver metastatic samples was measured at day 17 ($n = 5$ per group, two independent experiments). **g** Survival analysis of liver metastatic WT or $P2rx1^{-/-}$ mice within a duration of 5 weeks ($n = 10$ per group, two independent experiments). **h** Normal liver (D0) and two sequential stages (D3 and D17) of liver metastases in WT and $P2rx1^{-/-}$ mice were harvested for RNA-seq, and PCA analyses were performed ($n = 3$ for D0, $n = 4$ for D3 and D17). **i** Representative Ki67 immunohistochemical staining of WT and $P2rx1^{-/-}$ liver metastases at day 17 ($n = 4$ per group, two independent experiments). 100 μm of scale bar for low power fields, 25 μm of scale bar for high power fields. **j** Heatmap of immune checkpoint molecules in liver metastases of WT or $P2rx1^{-/-}$ mice at D3 and D17 ($n = 4$ per group). Bars represent mean ± standard deviation in (**d**, **f**). $P$ values are derived from two-sided Student's $t$ test (**d**, **f**), or log-rank test (**g**). Source data are provided as a Source data file.

distribution of P2RX1− and P2RX1+ neutrophils during PDAC liver metastasis, BM and peripheral blood (PB) from normal and liver metastatic mice were analyzed. Results showed that total neutrophils were significantly increased in the BM (Fig. 4a, b, e) and PB (Fig. 4c–e) of liver metastatic mice, indicating systemic granulopoiesis in the presence of liver metastasis. In addition, we found that frequency of P2RX1− neutrophils were increased in the BM (Fig. 4a, b, f) and PB (Fig. 4c, d, f) of PDAC liver metastatic mice, which suggested that P2RX1− neutrophils were substantially generated. CXCR4 is a key regulator of neutrophils released from BM[23]. By tethering neutrophils in BM, CXCR4 maintains neutrophil homeostasis in PB under both basal and stress granulopoiesis. We found that nearly all P2RX1+ and P2RX1− neutrophils in BM were CXCR4 + under steady conditions (Fig. 4a, b, g), as previously reported[23]. However, an apparent increase in the CXCR4− proportion of P2RX1− neutrophils was observed in liver metastatic mice, suggesting that P2RX1− neutrophils were more readily released from BM during liver metastasis.

Accumulating evidence has revealed the abundant heterogeneities of neutrophils[24]. Several classifications, including "mature or immature"[25], and "fresh or aged"[26] are adopted to characterize the neutrophil heterogeneity. CXCR2 is used to identify "mature" or "immature" neutrophils[25], and CXCR4 is used to identify "fresh" or "aged" neutrophils[26]. To further analysis the relationship between P2RX1 and the known neutrophil heterogeneity markers, we performed flow cytometry studies. In liver metastatic tissues, we found that almost all neutrophils were CXCR2+ (Supplementary Fig. 6a), which suggested that both P2RX1+ and P2RX1− neutrophils were mature neutrophils. Interestingly, we noticed that CXCR2 expression intensity was different between P2RX1+ and P2RX1− neutrophils, with P2RX1− neutrophil expressing higher level of CXCR2 (Supplementary Fig. 6a and b). Given that CXCR2 is an important chemokine receptor for regulation neutrophil migration, we suppose that higher CXCR2 may favor the enrichment of P2RX1− neutrophils in liver metastases. In addition, we observed that CXCR4 expression was different between P2RX1+ and P2RX1− neutrophils, with P2RX1− neutrophils expressing less CXCR4 (Supplementary Fig. 6c and d). It suggested that P2RX1− neutrophils were "fresher" than P2RX1+ neutrophils. Therefore, our results indicate that P2RX1− neutrophils in PDAC liver metastases are characterized with mature and fresh phenotypes.

Next, BM neutrophils were isolated from WT mice and stimulated with several typical stimulus, including G-CSF, GM-CSF, LPS, IFN-γ, and IL-4. RNA-seq was performed to determine the expression pattern of purinergic receptors. Results showed that $P2rx1$ was the second most highly expressed purinergic receptors in naïve BM neutrophils (Fig. 4h). Interestingly, we noticed that distinct from most purinergic receptors, all the stimulus could reduce the expression of $P2rx1$, which was further

verified by RT-qPCR (Supplementary Fig. 6e). It suggested that transcriptional susceptibility could contribute to the generation of P2RX1-negatively expressed neutrophils.

**Characterizing the immune and metabolic features of P2RX1-negative neutrophils in the microenvironment of PDAC liver metastases.** To better characterize the protumor nature of P2RX1− neutrophils, we compared the transcriptomes of P2RX1-deficient neutrophils purified from $P2rx1^{-/-}$ mice and P2RX1+ neutrophils purified from WT mice. Among 1153 significantly differentially expressed genes, 608 genes were upregulated, and 545 genes were downregulated in $P2rx1^{-/-}$ neutrophils (Supplementary Fig. 7a). KEGG analyses revealed that metabolism-related pathways were the most extensively enriched pathways in $P2rx1^{-/-}$ neutrophils (Fig. 5a). Signatures of mitochondria-associated metabolic pathways, including the TCA cycle and fatty acid metabolism, were upregulated in $P2rx1^{-/-}$ neutrophils, whereas signatures of glycolysis were enriched in P2RX1+ neutrophils. As mitochondrial metabolism frequently supports immunosuppressive phenotypes and glycolysis mainly characterizes immunostimulatory phenotypes[27], we then examined immune-associated gene expression in $P2rx1^{-/-}$ and P2RX1+ neutrophils. We observed that antitumoral N1-like neutrophil markers[28,29], including $TNF-\alpha$, $Fas$, and $Met$, were highly expressed in P2RX1+ neutrophils, whereas protumoral N2-like neutrophil markers[7,29–31], including $Arg1$, $Ccl5$, $Ccl17$, $Mif$, $Mmp9$, and $PD-L1$, were increased in $P2rx1^{-/-}$ neutrophils (Fig. 5b). RT-qPCR further verified that $TNF-\alpha$ expression was lower, whereas $PD-L1$ expression was higher in metastasis-infiltrated $P2rx1^{-/-}$ neutrophils when compared to P2RX1+ neutrophils (Supplementary Fig. 7b). In addition, $TNF-\alpha$ and $PD-L1$ expression were determined in P2RX1− neutrophils derived from WT mice. We found that expression patten of WT mice-derived P2RX1− neutrophils was similar to that of $P2rx1^{-/-}$ neutrophils (Supplementary Fig. 7b). These results identify the immunosuppressive phenotypes of liver metastatically infiltrated P2RX1-deficient neutrophils. Neutrophil extracellular traps (NETs) in promoting tumor metastasis have attracted much attention recently[22,32–34]. We observed that NETs were substantially generated in liver metastases, but no significant differences were seen between WT and $P2rx1^{-/-}$ mice (Supplementary Fig. 7c), indicating that P2RX1 might have no effect on NETs formation in PDAC liver metastasis.

Next, extracellular acid rate (ECAR) and oxygen consumption rate (OCR) were measured to quantify neutrophil glycolysis and mitochondrial oxidative phosphorylation (OXPHOS) in vitro. Tumor conditioned medium (TCM) from murine PDAC liver metastases was used to stimulate WT and $P2rx1^{-/-}$ neutrophils. We found that TCM significantly increased OCR in WT neutrophils, and which was further enhanced in $P2rx1^{-/-}$ neutrophils (Fig. 5c). In comparison, ECAR in WT neutrophils

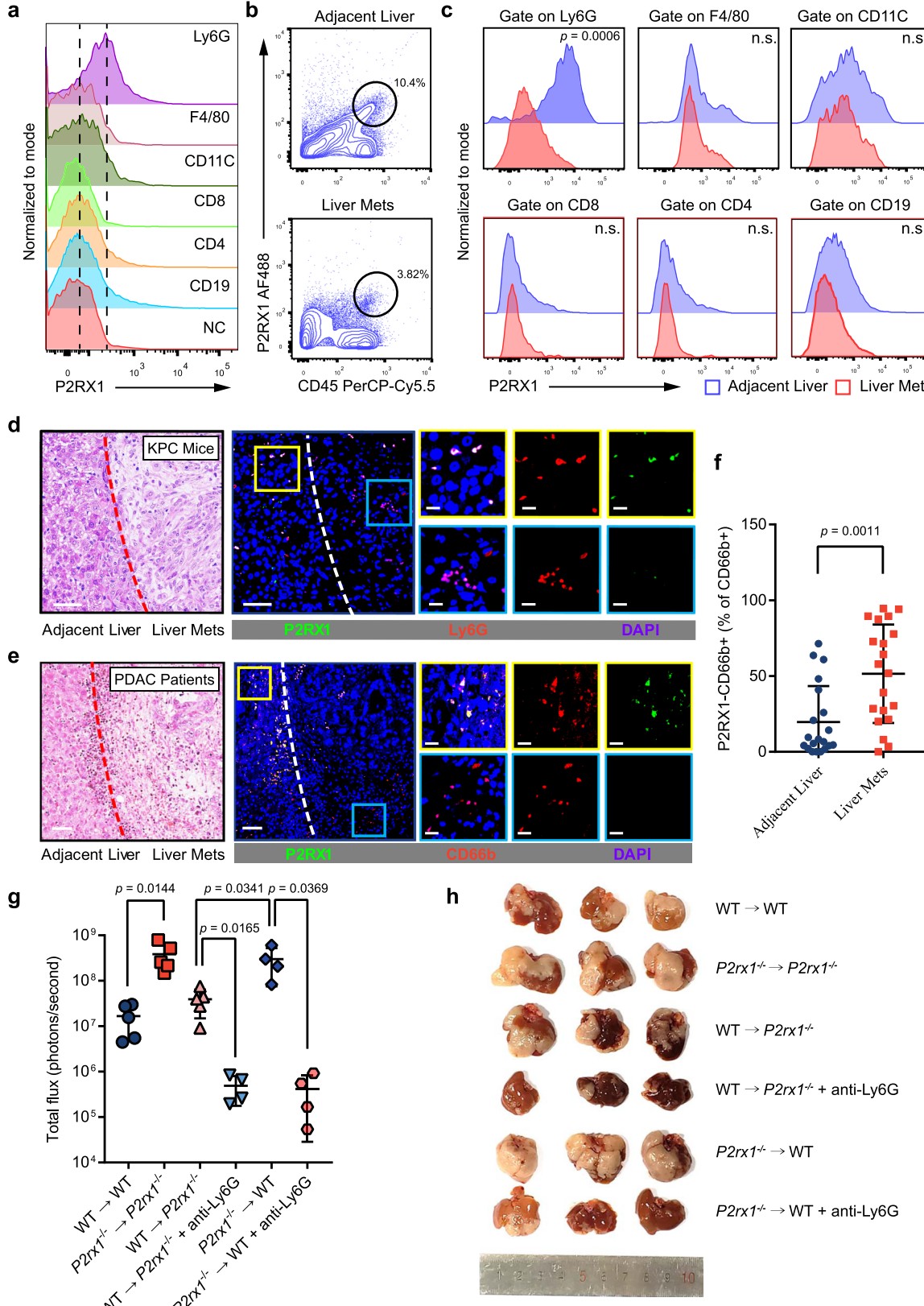

was mildly increased by TCM, and which was less obvious in $P2rx1^{-/-}$ neutrophils (Fig. 5c). In addition to the metabolic shift, we found that TCM could induce higher *PD-L1* expression in $P2rx1^{-/-}$ neutrophils, and lower *TNF-α* expression in $P2rx1^{-/-}$ neutrophils (Fig. 5d). Therefore, these in vitro results were similar

to in vivo RNA-sequencing results, indicating that neutrophil metabolic and immune phenotypes could be regulated by P2RX1.

Similar to macrophage polarization, recent evidence has indicated that neutrophils also harbor the ability to polarize to antitumoral N1-like neutrophils or protumoral N2-like

**Fig. 3 P2RX1− neutrophils accumulate in the microenvironment of PDAC liver metastases and promote metastatic tumor growth. a** Single cell suspension was obtained from mouse spleen and expression of P2RX1 was determined in indicated immune cell types (n = 3 per group, two independent experiments). Left dotted line indicates the mean of negative control (NC) (secondary antibody only), and right dotted line indicates the mean of Ly6G+ cells. **b** KPC cells were intrasplenically injected to seed livers of WT mice. A single cell suspension was obtained from liver metastases and adjacent liver tissues of WT mice at day 17. The frequency of CD45+P2RX1+ cells was determined by flow cytometry (n = 4 per group, three independent experiments). **c** KPC cells were intrasplenically injected to seed livers of WT mice. Immune cells were enriched from single cell suspension of liver metastases and adjacent liver tissues at day 17. P2RX1 expression in the indicated immune cell types was determined by flow cytometry (n = 4 per group, three independent experiments). **d** Representative images of H&E staining and immunofluorescence staining of P2RX1 (Green), Ly6G (Red) and DAPI (Blue) in KPC mice spontaneous liver metastases (representative results from six independent experiments). 50 μm scale bar for low power fields and 20 μm scale bar for high power fields. **e, f** H&E staining and immunofluorescence staining of P2RX1 (green), CD66b (red) and DAPI (blue) in a total of 20 clinical PDAC liver metastasis samples were performed. Representative images are shown in (**e**), and the percentages of P2RX1-CD66b+ cells are shown in (**f**). 50 μm scale bar for low power fields and 20 μm scale bar for high power fields. **g, h** KPC cells were intrasplenically injected to seed livers of BM chimeras: WT → WT, $P2rx1^{-/-}$ → $P2rx1^{-/-}$, WT → $P2rx1^{-/-}$, and $P2rx1^{-/-}$ → WT. Neutrophils were depleted in WT → $P2rx1^{-/-}$ and $P2rx1^{-/-}$ → WT mice by intraperitoneal injection of anti-Ly6G (clone 1A8) antibody. At day 17, liver metastases were analyzed by in vivo imaging (**g**, n = 5 for WT → WT, $P2rx1^{-/-}$ → $P2rx1^{-/-}$ and WT → $P2rx1^{-/-}$ groups, n = 4 for WT → $P2rx1^{-/-}$ + anti-Ly6G, $P2rx1^{-/-}$ → WT and $P2rx1^{-/-}$ → WT + anti-Ly6G groups, two independent experiments), and representative images of liver metastatic samples were shown (**h**). Bars represent mean ± standard deviation in (**f, g**). P values are derived from two-sided Student's t test (**c, f**), or one-way ANOVA and Tukey's multiple comparisons test (**g**). Source data are provided as a Source data file.

neutrophils[29]. Our RNA-seq results confirmed that the proinflammatory mediators LPS and IFN-γ induced antitumoral N1-like neutrophil gene expression, whereas the anti-inflammatory mediator IL-4 induced protumoral N2-like neutrophil gene expression (Supplementary Fig. 7d). However, the detailed mechanisms remain far less understood. Given that metabolic reprogramming is crucial for determining immune responses in immune cells, ECAR and OCR were then measured to quantify neutrophil metabolic programming during polarization. The results showed that LPS + IFN-γ-polarized WT neutrophils had a significantly increased ECAR, whereas IL-4-polarized WT neutrophils had an apparently enhanced OCR (Supplementary Fig. 7e and f). Compared to WT neutrophils, $P2rx1^{-/-}$ neutrophils had a similar baseline ECAR and OCR. However, unlike WT neutrophils, we observed that the increased ECAR was impaired in $P2rx1^{-/-}$ neutrophils when the cells were polarized with LPS + IFN-γ, whereas the OCR was further increased when $P2rx1^{-/-}$ neutrophils were polarized with IL-4 (Supplementary Fig. 7e and f). This result indicated that $P2rx1^{-/-}$ neutrophils were less glycolytic but more oxidative. A nonhydrolyzed ATP analog, ATP-γ-S, promoted the ECAR in LPS + IFN-γ-polarized WT neutrophils and inhibited the OCR in IL-4-polarized WT neutrophils (Supplementary Fig. 7e and f). However, no significant influence was shown in $P2rx1^{-/-}$ neutrophils. These findings further confirmed that deficiency of P2RX1 signaling accounted for the enhanced neutrophil mitochondrial OXPHOS and reduced glycolysis. Adequate mitochondrial mass is essential for OXPHOS. By detecting MitoTracker Green staining and mitochondrial DNA, we found that N1- and N2-polarized $P2rx1^{-/-}$ neutrophils had increased mitochondrial mass compared to that of WT neutrophils (Supplementary Fig. 7g and h).

**Upregulated Nrf2 is essential for shaping the immunosuppressive phenotypes of P2RX1-negative neutrophils.** Nuclear factor-erythroid 2 p45-related factor 2 (Nrf2, gene name Nfe2l2) is a master transcription factor that regulates diverse cellular responses to environmental stresses[35]. KEGG pathway analysis indicated that Nrf2-related pathways, including carbon monoxide metabolism, purine metabolism, the TCA cycle and fatty acid metabolism[36], were enhanced in $P2rx1^{-/-}$ neutrophils (Fig. 5a). Moreover, GSEA revealed that Nrf2 target genes were significantly enriched in $P2rx1^{-/-}$ neutrophils (Fig. 6a), based on mouse Nrf2 ChIP-sequencing data sets from the ChEA database[37]. RT-qPCR and flow cytometry were performed and confirmed that Nrf2 expression was increased in liver metastatic

$P2rx1^{-/-}$ neutrophils (Fig. 6b and Supplementary Fig. 8a, b). In addition, nuclear Nrf2 accumulation was also reinforced in $P2rx1^{-/-}$ neutrophils, indicating higher Nrf2 activity (Fig. 6c). Nrf2 is known as an important anti-inflammatory transcription factor that induces metabolically reprogrammed immune-tolerant macrophages and DCs[38,39]. However, the specific role of Nrf2 in shaping the neutrophil immune phenotype remains largely unknown. Therefore, we hypothesized that the reprogrammed metabolic and immune phenotype in $P2rx1^{-/-}$ neutrophils may be due to enhanced Nrf2 activation.

Flow cytometry analysis showed that several stimuli, including GM-CSF, LPS, IFN-γ, LPS + IFN-γ and TCM-induced increased Nrf2 expression in $P2rx1^{-/-}$ neutrophils (Supplementary Fig. 8c). Next, the effect of Nrf2 on neutrophil metabolism was determined with the Nrf2 inhibitor ML385. The Nrf2 inhibitor had little effect on the baseline ECAR and OCR of WT or $P2rx1^{-/-}$ neutrophils (Fig. 6d, e). However, the ECAR of N1-polarized $P2rx1^{-/-}$ neutrophils was markedly increased, whereas the OCR was significantly increased in N2-polarized $P2rx1^{-/-}$ neutrophils compared with those of WT neutrophils (Fig. 6d, e). This result indicated that increased Nrf2 activity in $P2rx1^{-/-}$ neutrophils contributed to metabolic reprogramming during neutrophil polarization. Next, we determined the anti-inflammatory effects of Nrf2. The results showed that IL-1α and TNF-α were further elevated in proinflammatory N1-polarized $P2rx1^{-/-}$ neutrophils compared to those of WT neutrophils (Fig. 6f). Considering that enhanced mitochondrial OXPHOS predominantly supports an immunosuppressive phenotype, while increased glycolysis frequently supports an immune-promoting phenotype[40], we considered that Nrf2 was critical for shaping immune-tolerant neutrophils.

**P2RX1-negative neutrophils facilitate the immunosuppressive microenvironment in PDAC liver metastases by upregulating PD-L1 expression.** RNA-seq evidence from clinical and murine PDAC liver metastatic tissues showed that reduced P2RX1 expression correlated with immune exhaustion-associated genes (Figs. 1e, 2f), and purified $P2rx1^{-/-}$ neutrophils had increased PD-L1 expression (Fig. 5b). Next, we examined the expression of the immune exhaustion PD-1 and CTLA-4 signaling axes in liver metastasis. The results showed that CD8+ T cell-expressed PD-1 and CTLA-4 were upregulated in liver metastases of WT mice and further increased in $P2rx1^{-/-}$ mice (Fig. 7a and Supplementary Fig. 9a). In parallel, neutrophil-expressed PD-L1 was significantly upregulated in infiltrating P2RX1− neutrophils in

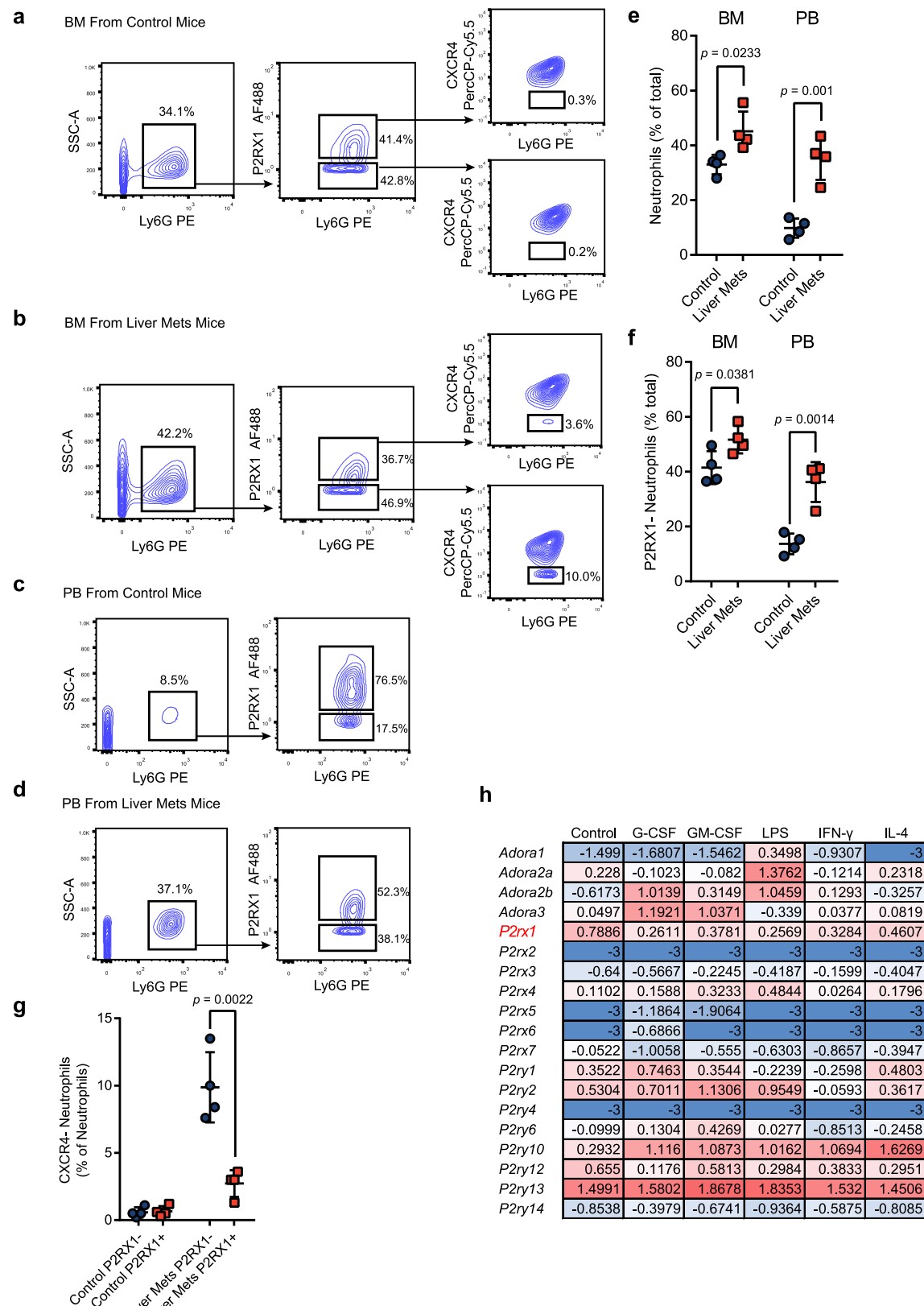

**Fig. 4 PDAC liver metastasis systematically mobilizes and recruits P2RX1− neutrophils. a–g** KPC cell was intrasplenically injected to seed livers of WT mice. Bone marrow (BM) and peripheral blood (PB) were obtained at day 17. Frequencies of neutrophils, P2RX1− neutrophils, and CXCR4+ neutrophils were determined by flow cytometry and quantitative results were shown ($n = 4$ per group, three independent experiments). **h** Bone marrow neutrophils were isolated from WT mice and stimulated with indicated stimulus. RNA-seq were performed and $\text{Log}_2$ (FPKM+0.001) value of purinergic receptors were shown ($n = 1$ per group). Bars represent mean ± standard deviation in (**e–g**). $P$ values are derived from two-sided Student's $t$ test (**e–g**). Source data are provided as a Source data file.

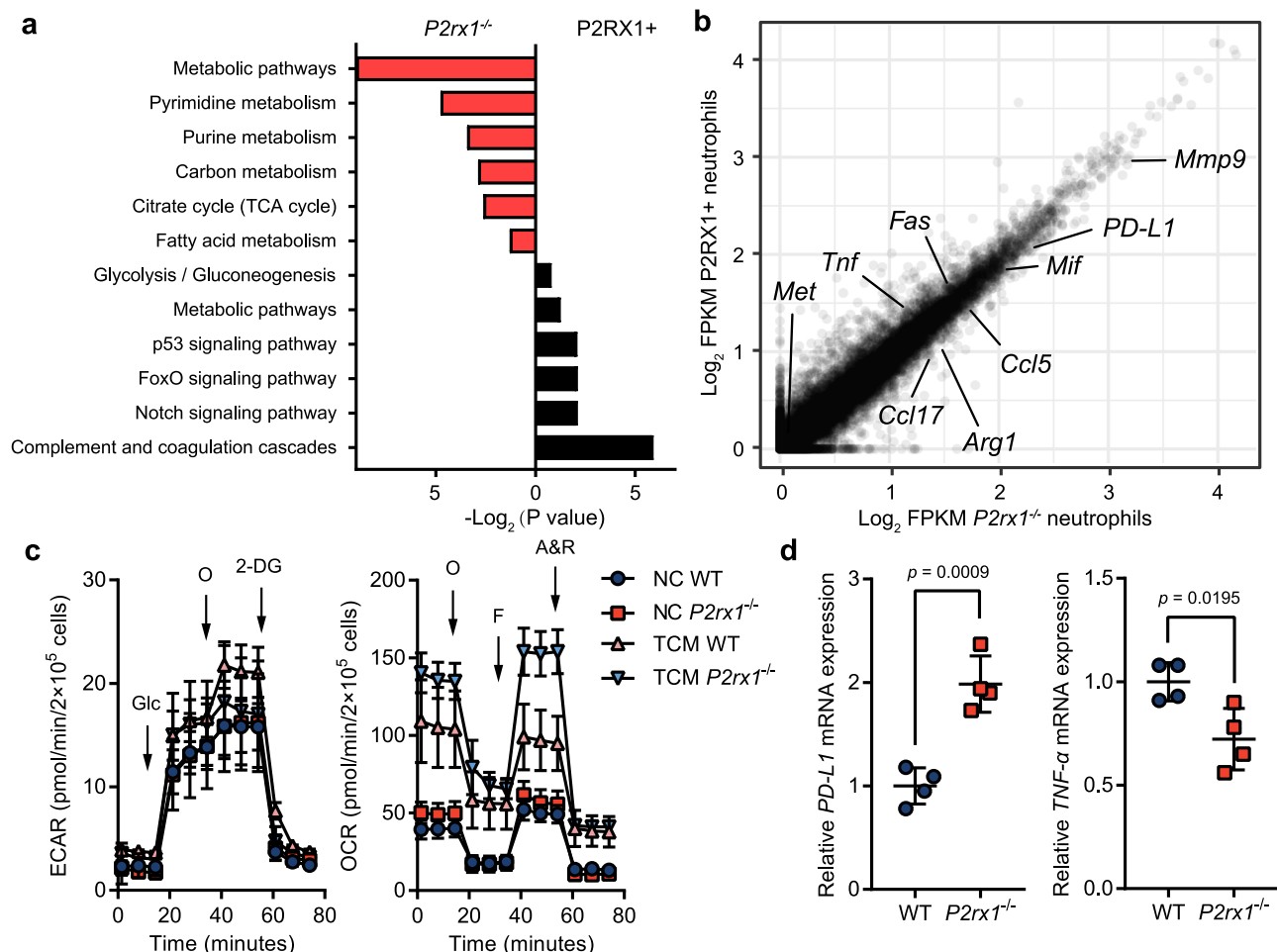

**Fig. 5 Characterizing the immune and metabolic features of P2RX1− neutrophils in the microenvironment of PDAC liver metastases. a**, **b** KPC cells were intrasplenically injected to seed livers of WT and $P2rx1^{-/-}$ mice. A single cell suspension was obtained from liver metastases at day 17. Then, P2RX1+ neutrophils were purified from WT mice, and $P2rx1^{-/-}$ neutrophils were purified from $P2rx1^{-/-}$ mice for RNA sequencing. The results of KEGG analysis are shown in (**a**), and comparative expression of genes is shown in (**b**) ($n = 1$ per group). **c** Bone marrow neutrophils were isolated from WT and $P2rx1^{-/-}$ mice and stimulated with tumor conditioned medium (TCM). The ECAR and OCR were then measured by a Seahorse assay in (**c**), and PD-L1 and TNF-α were detected by RT-qPCR in (**d**) ($n = 4$ per group, two independent experiments). Glc, glucose; O (ECAR), oligomycin; 2-DG, 2-deoxyglucose; O (OCR), oligomycin; F, FCCP (carbonyl cyanide 4-[trifluoromethoxy] phenylhydrazone); A & R, antimycin A and rotenone. Bars represent mean ± standard deviation in (**c**, **d**). P values are derived from two-sided Student's t test (**d**). Source data are provided as a Source data file.

WT mice and in infiltrating P2RX1-deficient neutrophils in $P2rx1^{-/-}$ mice (Fig. 7a and Supplementary Fig. 9a). CD80 but not CD86 or PD-L2 expression was also upregulated in neutrophils (Supplementary Fig. 9a). However, the increase in CD80+ neutrophils was much less than that of PD-L1+ neutrophils (Supplementary Fig. 9a). Human PDAC tumor cells have been reported to express PD-L1, but there exists abundant variations between different cohorts[41,42]. Different KPC cell strains have also been reported to express heterogeneous PD-L1[43]. In the present study, flow cytometry and immunofluorescence studies showed that PD-L1 was mainly expressed in Ly6G+ neutrophils but not KPC cells (Supplementary Fig. 9b, c). Therefore, we suggest that neutrophil-derived PD-L1 might be an essential contributor to PD-1-induced CD8+ T cell exhaustion in liver metastasis.

To explore the mechanism of PD-L1 expression, TCM was used to treat WT or $P2rx1^{-/-}$ neutrophils. We observed that PD-L1 expression was significantly increased in both groups, and $P2rx1^{-/-}$ neutrophils had a higher PD-L1 expression level than WT neutrophils (Fig. 7b and Supplementary Fig. 9d). Next, inflammation-associated cytokines and chemokines in murine

PDAC liver metastases were screened to determine the specific metastasis-derived factors that promoted neutrophil PD-L1 expression. We found that *Csf2* (*GM-CSF*) was the most significantly upregulated cytokine (Supplementary Fig. 9e) and that *GM-CSF* expression was closely related to *PD-L1* expression (Supplementary Fig. 9f). In vitro GM-CSF treatment alone induces higher PD-L1 expression in $P2rx1^{-/-}$ neutrophils than in WT neutrophils (Fig. 7b and Supplementary Fig. 9g). In addition, antibody blockade of GM-CSF significantly abolished TCM-induced PD-L1 expression on neutrophils (Fig. 7b and Supplementary Fig. 9d). Autocrine-derived ATP is sufficient to initiate neutrophil P2RX1 signaling. Using apyrase to hydrolyze extracellular ATP, PD-L1 expression was further increased in GM-CSF-stimulated WT but not $P2rx1^{-/-}$ neutrophils (Supplementary Fig. 9g). In addition, ATP-γ-S inhibited neutrophil PD-L1 expression in WT but not $P2rx1^{-/-}$ neutrophils (Supplementary Fig. 9g). These results indicated that P2RX1-deficient neutrophils had increased PD-L1 expression in the presence of the PDAC metastasis-related factor GM-CSF.

We noticed that PD-L1 and the anti-inflammatory transcription factor Nrf2 were both upregulated in liver metastatic

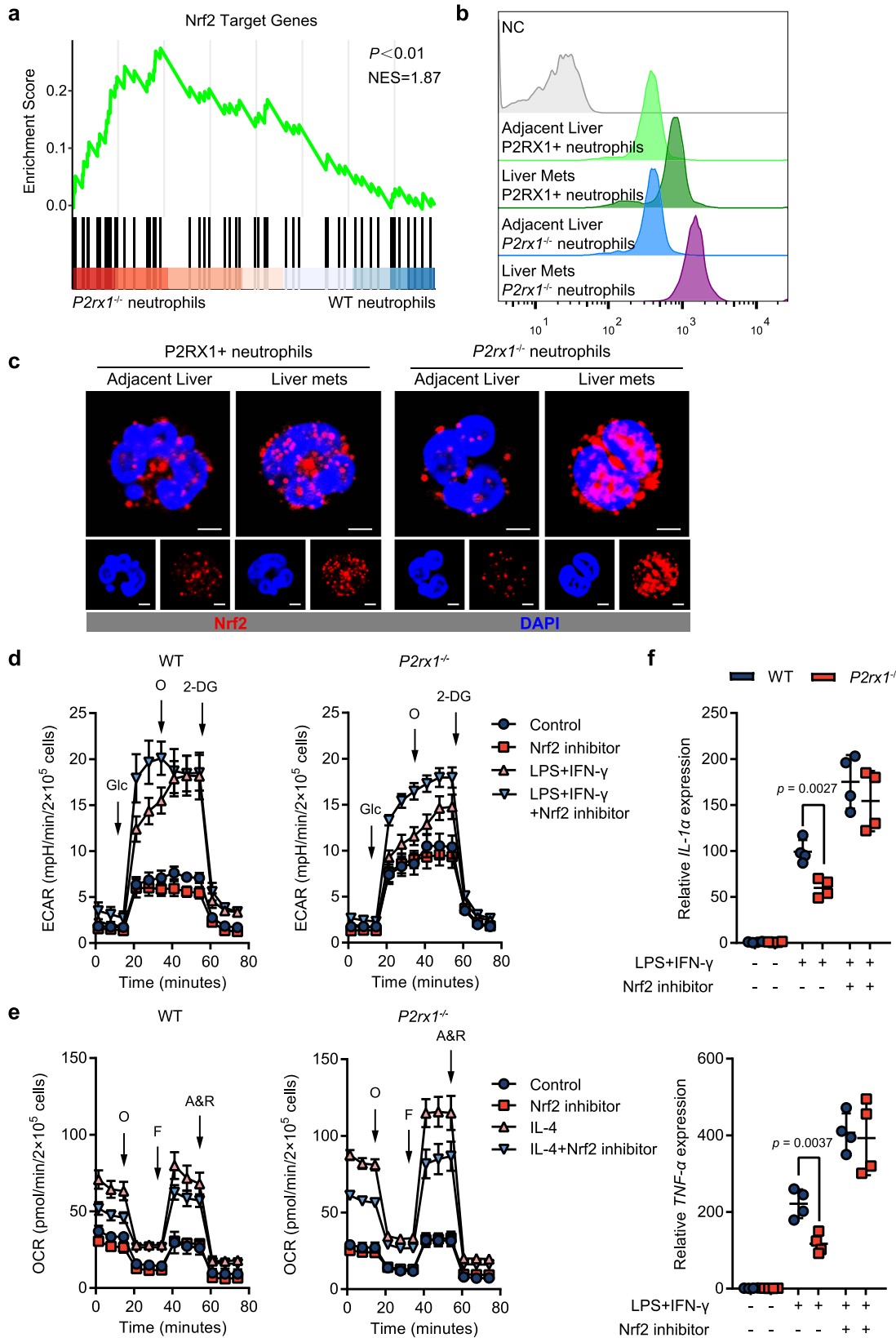

$P2rx1^{-/-}$ neutrophils in vivo and that TCM or GM-CSF stimulated $P2rx1^{-/-}$ neutrophils in vitro. However, whether Nrf2 regulated neutrophil PD-L1 expression was unclear. Using a Nrf2 inhibitor, we found that TCM- and GM-CSF-induced PD-L1 expression was significantly reduced, suggesting that Nrf2 regulated PD-L1 expression (Fig. 7b, Supplementary Fig. 9d and

g). Next, ChIP-PCR analysis was performed to determine the direct regulation of PD-L1 expression by Nrf2. Two peaks of the PD-L1 gene that might be mediated by Nrf2 were predicted using the Integrative Genomics Viewer (IGV) (Fig. 7c). Using specific primers for the two peaks, we observed that GM-CSF stimulation significantly promoted Nrf2 binding to the PD-L1 gene in peak 1

**Fig. 6 Upregulated Nrf2 is essential for shaping the immune-tolerant phenotype of P2RX1− neutrophils. a** Gene set enrichment analysis comparing RNA-seq data of P2RX1+ neutrophils and *P2rx1*−/− neutrophils based on Nrf2 target genes. The *p* value and normalized enrichment score (NES) were shown. **b, c** Single-cell suspensions were obtained from liver metastases of WT and *P2rx1*−/− mice at day 17. Intracellular Nrf2 was stained and detected by flow cytometry (**b**) or laser scanning confocal microscopy (**c**) in WT P2RX1+ neutrophils and *P2rx1*−/− neutrophils (*n* = 4 per group, three independent experiments). The scale bar is 2.5 μm. **d** Bone marrow neutrophils were isolated from WT and *P2rx1*−/− mice and stimulated with LPS + IFN-γ in the presence or absence of a Nrf2 inhibitor. The ECAR was then measured by a Seahorse assay (*n* = 4 per group, two independent experiments). Glc, glucose; O, oligomycin; 2-DG, 2-deoxyglucose. **e** Bone marrow neutrophils were isolated from WT and *P2rx1*−/− mice and stimulated with IL-4 in the presence or absence of a Nrf2 inhibitor. The OCR was then measured by a Seahorse assay (*n* = 4 per group, two independent experiments). O, oligomycin; F, FCCP (carbonyl cyanide 4-[trifluoromethoxy] phenylhydrazone); A & R, antimycin A and rotenone. **f** Bone marrow neutrophils were isolated from WT or *P2rx1*−/− mice and stimulated with LPS + IFN-γ and an inhibitor Nrf2 inhibitor. IL-1β and TNF-α were determined by RT-qPCR (*n* = 4 per group, three independent experiments). Bars represent mean ± standard deviation in (**d**–**f**). *P* values are derived from permutation test (**a**), two-sided Student's *t* test (**f**). Source data are provided as a Source data file.

located in the promoter but not in peak 2 or other negative controls (Fig. 7c). To further confirm that Nrf2 could directly bind to *PD-L1* promotor and regulate *PD-L1* transcription, we performed luciferase reporter assay. We cloned WT *PD-L1* promoter into pGLB3 luciferase reporter vector, and the results showed that co-transfection of Nrf2 markedly increased the promotor activity in NIH3T3 cells (Supplementary Fig. 10a). In contrast, mutation of *PD-L1* at Peak1 did not response to Nrf2 co-transfection (Supplementary Fig. 10a). These results suggest that *PD-L1* transcription is directly regulated by Nrf2.

Nrf2 is a ROS-sensitive transcription factor[44]. We observed that P2RX1-deficient neutrophils generated increased intracellular ROS when activated by GM-CSF (Fig. 7d and Supplementary Fig. 10b), which was consistent with the previous reports[45]. Interestingly, stimulations (LPS and IFN-γ) that induced robust ROS generation and Nrf2 expression also promoted neutrophil PD-L1 expression (Supplementary Fig. 10b and c). However, the cytokines (M-CSF and IL-10) that had little effect on ROS generation or Nrf2 expression also failed to promote neutrophil PD-L1 expression (Supplementary Fig. 10b and c). Using N-acetylcysteine (NAC) to reduce intracellular ROS, we found that GM-CSF-, LPS- and IFN-γ-induced PD-L1 expression was significantly abolished in *P2rx1*−/− neutrophils (Supplementary Fig. 10c). These results suggest that increased intracellular ROS in *P2rx1*−/− neutrophils may favor the elevated PD-L1 expression by activating the ROS-sensitive transcription factor Nrf2.

To further address the direct effect of neutrophil-expressed PD-L1 on CD8+ T cell exhaustion, cytotoxic T lymphocyte (CTL) proliferation experiments were performed in vitro. Ovalbumin (OVA)-specific TCR transgenic OT-1 cells were isolated from OT-1 mice, and the OVA–derived peptide SIINFEKL was used to generate CTLs. We observed that the proliferation of CTLs was markedly inhibited by GM-CSF-primed *P2rx1*−/− neutrophils, which was then reversed by anti-PD-1 neutralizing antibody (Fig. 7e and Supplementary Fig. 10d). Next, in vitro cytotoxicity studies were performed. We generated OVA-expressed KPC cells (KPC-OVA) as target cells with lentiviral vector (LV)-OVA, and empty LV was transfected as a control. We found that OVA-specific CTLs were less cytotoxic to KPC-OVA cells in the presence of GM-CSF-primed *P2rx1*−/− neutrophils than in the presence of WT neutrophils (Fig. 7f). Anti-PD-1 antibody markedly improved the cytotoxicity of OVA-specific CTLs. In addition, similar results of CTL proliferation and cytotoxicity were observed when an anti-PD-L1 neutralizing antibody was used (Supplementary Fig. 10e and f). These results suggest that *P2rx1*−/− neutrophils facilitate the immune evasion of PDAC in a PD-L1/PD-1-dependent manner.

## Discussion
Targeting the immunosuppressive environment has been shown to be a promising therapeutic strategy for treating many types of cancers. However, the efficacy of immunotherapies is limited in primary PDAC due to the desmoplastic and nonimmunogenic tumor microenvironment that lacks CTLs infiltration[46]. Recent studies showed that liver metastatic PDAC contains certain amounts of exhausted tumoricidal CTLs[6], suggesting that targeting the immunosuppressive microenvironment could be a promising strategy for treating liver metastasis of PDAC[47]. Neutrophils are newly recognized crucial contributors to tumor metastases[21]. However, the role of neutrophils in tumor progression remains controversial[8]. Both antitumoral and protumoral neutrophil subsets have been identified, and relatively little is known about how heterogeneity is shaped. Purinergic signaling is crucial for regulating neutrophil immune functions[48]. A recent study revealed that in the early-stage of cancer, bone marrow neutrophils ATP production was upregulated and autocrine ATP-involved purinergic signaling favored neutrophil spontaneous migration[49]. However, in the late-stage of cancer, ATP production and migratory activity were indistinguishable, and the neutrophils displayed potent immunosuppressive activity[49]. This evidence indicates that varied purinergic signaling may lead to heterogeneous neutrophil functions in tumor progression[49]. In the present study, we identified that a subset of neutrophils that deficient of purinergic receptor P2RX1 were systemically accumulated in the presence of PDAC liver metastasis. P2RX1− neutrophils displayed immunosuppressive phenotypes and inhibited PDAC-specific CTLs by expressing high level PD-L1.

By regulating transcriptional network of antioxidant, anti-inflammatory and metabolism, Nrf2 acts as a master transcription factor that controls adaptive responses to various environmental stresses[35]. In a severe infection murine model, Nrf2 was found to restrain nuclear factor-κB-involved innate immune responses in myeloid macrophages and neutrophils[50]. Our previous study observed that autocrine-induced activation of P2RX1 favored bacterial LPS-induced neutrophil activation[13]. However, the relation between Nrf2 and P2RX1 in neutrophils is unclear. In the present study, we showed that Nrf2 activity was upregulated in PDAC liver metastasis-infiltrated P2RX1− neutrophils. As a redox-sensitive transcription factor, we suggested that elevated ROS generation in P2RX1− neutrophil might contribute to the enhanced Nrf2 activity. Using chemical antagonist to block Nrf2, we observed that immunosuppressive phenotypes, including upregulated *PD-L1* expression and suppressed *TNF-α* expression were significantly restored in P2RX1− neutrophils. Therefore, the present study reveals that neutrophil Nrf2 activity can be elevated as a consequence of P2RX1 deficiency, and enhanced Nrf2 in P2RX1− neutrophils is essential for mediating suppression on antitumor immune responses in PDAC liver metastasis.

Accumulating evidence has shown that altered metabolic profiles characterize and impact immune cell fate[51]. RNA-seq results revealed that metabolic and immune phenotypes are significantly varied between the two neutrophil subsets with differential P2RX1 expression, with P2RX1-deficient neutrophils highly

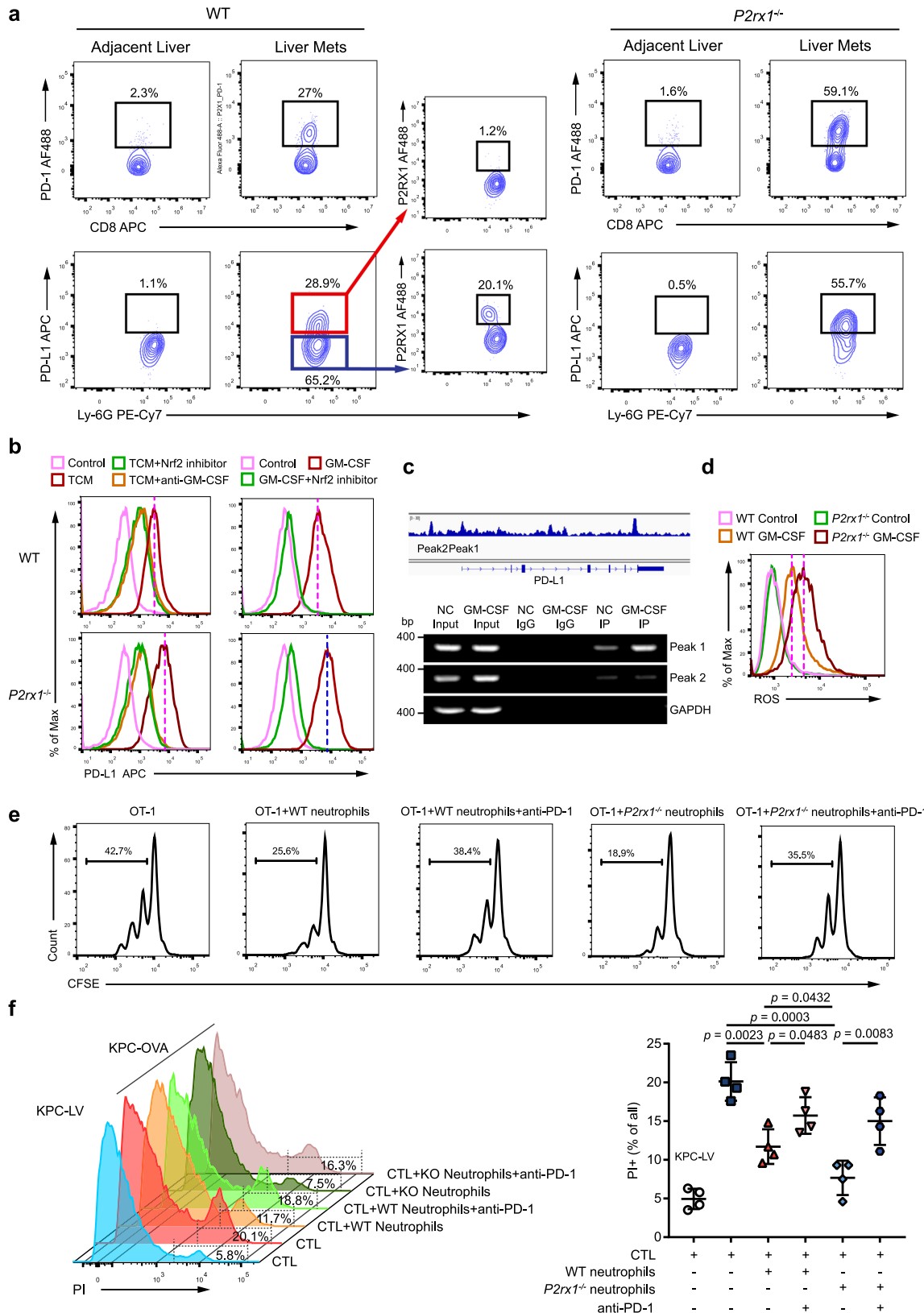

expressing protumoral N2-like neutrophil markers, including *Arg1*, *PD-L1,* and *Ccl5*, and mitochondrial metabolism-related genes. Functional polarization of neutrophils into N1-like or N2-like cells in vitro is a simplified classification, and this schema provides an operationally useful guide to study neutrophil plasticity in vitro. Similar to the polarization of macrophages,

metabolic reprogramming was also observed in polarized neutrophils. We found that $P2rx1^{-/-}$ neutrophils were less glycolytic but more oxidative. Further mechanistic studies revealed that Nrf2 accounts for metabolic reprogramming and suppressed proinflammatory cytokine expression. Considering that glycolysis frequently supports immunostimulatory responses, whereas

**Fig. 7 P2RX1− neutrophils facilitate an immunosuppressive microenvironment in PDAC liver metastases by upregulating PD-L1 expression. a** A single cell suspension was obtained from liver metastases of WT and *P2rx1*−/− mice at day 17. Flow cytometry was performed to detect the frequency of CD8 +PD-1+ (upper) and Ly6G+PD-L1+ or Ly6G+PD-L1− (lower) cells. P2RX1 expression was further determined in Ly6G+PD-L1+ and Ly6G+PD-L1− cells from WT mice (*n* = 4 per group, three independent experiments). **b** Bone marrow neutrophils were isolated from WT and *P2rx1*−/− mice and stimulated with tumor conditioned medium (TCM) or GM-CSF in the presence of a Nrf2 inhibitor or anti-GM-CSF neutralizing antibody. PD-L1 expression was detected by flow cytometry (*n* = 4 per group, three independent experiments). **c** Integrative Genomics Viewer (IGV) was used to predict the two peaks that PD-L1 gene might be mediated by Nrf2 (upper). Bone marrow neutrophils were isolated from WT and *P2rx1*−/− mice and stimulated with GM-CSF. Binding of PD-L1 gene by Nrf2 was detected by ChIP-PCR (*n* = 3 per group, two independent experiments). **d** Bone marrow neutrophils were isolated from WT and *P2rx1*−/− mice and stimulated with GM-CSF. Intracellular ROS was detected by flow cytometry (*n* = 4 per group, three independent experiments). **e** Antigen activated CTLs was co-cultured with GM-CSF primed WT or *P2rx1*−/− neutrophils. Cell proliferation was analyzed with CSFE staining in the presence or absence of anti-PD-1 neutralizing antibody (*n* = 4 per group, three independent experiments). **f** KPC cells were transfected with empty lentiviral vector (KPC-LV) or OVA (KPC-OVA). Antigen-activated CTLs were co-cultured with KPC-LV or KPC-OVA cells in the presence of GM-CSF-primed WT or *P2rx1*−/− neutrophils. Cytotoxicity was determined in the presence or absence of anti-PD-1 neutralizing antibody by counting the number of PI+ cells (*n* = 4 per group, three independent experiments). Bars represent mean ± standard deviation in (**f**). *P* values are derived from one-way ANOVA and Tukey's multiple comparisons test (**f**). Source data are provided as a Source data file.

mitochondrial OXPHOS frequently supports immunosuppressive responses, we suggest that Nrf2-supported mitochondrial metabolism in P2RX1-deficient neutrophils may be associated with the shaping of their immunosuppressive phenotypes.

We noticed that autocrine-induced P2RX1 activation was reported to promote CD4+ T cells activation and the mechanism seemed to be associated with upregulated mitochondrial activity[52,53]. Using P2RX1 antagonist to treat CD4+ T cells for 20 min, the authors observed that P2RX1-evoked cytosolic $Ca^{2+}$ response, and mitochondrial $Ca^{2+}$ uptake were significantly reduced. In line with these, our previous study also observed that autocrine-induced P2RX1 activation promoted neutrophil $Ca^{2+}$ influx, and increased cytosolic $Ca^{2+}$ subsequently reinforces neutrophils' responses to LPS within 10 min[13]. It is noticed that the physiological accumulation of cytosolic $Ca^{2+}$ is able to support mitochondrial activity[54]. However, a long term of $Ca^{2+}$ overload is harmful to mitochondrial activity directly[55], and dysregulated $Ca^{2+}$ influx may reduce mitochondrial activity indirectly by inhibiting Nrf2[56]. Overall, studies from our group and others reveal the role of autocrine-activated P2RX1 activation in promoting neutrophils and CD4+ T cells immune responses, and different mechanisms may be involved in divergent conditions.

As one of the major biochemical constituents of the tumor microenvironment, extracellular ATP plays crucial roles in tumor progression by acting on both tumor and host cell-expressed P2 purinergic receptors[11]. Studies from our group and others have demonstrated that tumor cells upregulate the P2 purinergic receptors P2RY1, P2RY2, P2RX3 and P2RX7 and utilize extracellular ATP to support tumor growth and metastasis[11,57]. However, in immune cells, extracellular ATP predominantly facilitates antitumor immunity[58]. Evidence from our group and others showed that extracellular ATP amplifies T cell receptor (TCR) signaling in lymphocytes and induces proinflammatory activation of macrophages, DCs and neutrophils[13,48,58]. Therefore, why dramatically accumulated extracellular ATP fails to give rise to adequate antitumor immune responses is unclear. In the present study, we identified that liver metastatic PDAC evades antitumor immunity by mobilizing and recruiting P2RX1− neutrophils. Despite the presence of extracellular ATP, reduced P2RX1 expression may reprogram the immunosuppressive features of this subset of neutrophils. Give that heterogeneous expression of purinergic receptors is also reported in other immune cell types[59], we speculate that the strategy to accumulate certain purinergic receptor-positive or -negative immune cell subtype may be adopted in other pathological conditions as well.

In conclusion, our studies uncovered a mechanism by which metastatic PDAC tumors evade antitumor immunity by

accumulating a subset of immunosuppressive P2RX1-negative neutrophils in an ATP-rich microenvironment. Future immunotherapies aimed at this pathological subset of neutrophils may help to develop potential strategies for treating PDAC liver metastases. However, the specific origin, manner of recruitment and clinical relevance remain to be determined and need more study.

## Methods

**Metastatic animal models**. All animal experiments were undertaken in accordance with the National Institutes of Health Guide for the Care and Use of Laboratory Animals. With the help from GemPharmatech Co., Ltd., P2RX1 knockout (*P2rx1*−/−) mice were constructed. Guide RNA sequences targeting Exon2–Exon3 of P2rx1 were: S1: CATCCAACGACGCAAGTGGC (PAM: TGG); S2: TGGTTCGCAAGTAGTTTCCC (PAM: AGG); S3: ATGGTGCTGTTGCG GGGCAC (PAM: TGG); S4: TTGGTCCTTGGTGGTAATGT (PAM: GGG). Homozygous *P2rx1*−/− mice and WT littermates were always generated from heterozygous parents. Following primers were used for genotyping WT, *P2rx1*−/− and heterozygous mice: forward: CACAGCCTTTGCTAGTGCCA, reverse: AGTGCAGCCACTGTCATCTT. No obvious differences of selective fertilization/ mortality in utero, development, and growth were observed between *P2rx1*−/− and WT littermates. We found that genetic ablation of P2RX1 resulted in male infertility, which was consistent with a previous report indicating that P2RX1 was involved in neurogenic vas deferens contraction[12]. $Kras^{G12D/+}$;$Trp53^{R172H/+}$;$Pdx$-$1$-Cre (KPC) mice were purchased from The Jackson Laboratory, and 20-26 weeks male or female KPC mice were used. KPC mice-derived PDAC cell line FC1199, referred to as KPC, was a generous gift from Professor Jing Xue (State Key Laboratory of Oncogenes and Related Genes, Shanghai Cancer Institute, Ren Ji Hospital, School of Medicine, Shanghai Jiao Tong University). Spontaneous liver metastatic PDAC tumors were obtained from KPC mice. Experimental liver metastasis was performed by intrasplenic injection of firefly luciferase transfected $1 × 10^6$ KPC cells in to 6–8 weeks male WT or *P2rx1*−/− mice. At the indicated time points, luciferin emission imaging was measured using the IVIS Spectrum with Perkin Elmer LifeImage 4.5.5 software to assay metastatic burden. For neutrophil depletion, daily 12.5 μg of anti-Ly6G antibodies (Biolegend) were intraperitoneally administered 1 day before or 4 days after KPC cells injection. More than 90% neutrophils in blood and liver could be removed based on our preliminary experiments. Animals were housed in East China Normal University specific pathogen free (SPF) animal facility, in temperatures 20–22 °C, humidity 30–70% and a 12-h light/12-h dark cycle.

**Mouse cells isolation and flow cytometry**. Mice were euthanized and liver metastases were harvested. Tissues were carefully minced and digested with 2 mg/ mL collagenase A (Sigma) and 1× DNase I (Sigma) for 30 min. Digestion was quenched by fetal bovine serum (FBS) and filtered with 70 μm Nylon mesh. After $600 × g$ centrifuging for 5 min, cells were resuspended in staining buffer (2%FBS in PBS). For enriching immunocytes in liver metastases, cells were centrifuged through a 37.5% Percoll at $850 × g$ for 35 min. Spleen was minced and triturated through 40 mm filters. Bone marrow was flushed with PBS and triturated through 70 mm filters. Bone marrow neutrophils were obtained by centrifugation with a discontinuous Percoll gradient (78%, 69%, and 52%) and harvested between the 78% and 69% layers. RPMI1640 containing 10% heated inactivated FBS was applied as cell culture medium.

Single cell suspensions were blocked with anti-mouse CD16/CD32 antibodies (Thermo Fisher) for 10 min and labeled with indicated fluorophore-conjugated antibodies PerCP/Cyanine5.5 anti-mouse CD45 (eBioscience, 1:100), PE anti-

mouse Ly6G (Biolegend, 1:100), PE/Cy7 anti-mouse Ly6G (Biolegend, 1:100), FITC anti-mouse PD-1 (eBioscience, 1:100), APC anti-mouse PD-L1 (Biolegend, 1:100), PE anti-mouse PD-L2 (Biolegend, 1:100), PE anti-mouse CD80 (BD, 1:100), PE anti-mouse CD86 (BD, 1:100), Alex Fluor 488 anti-mouse F4/80 (eBioscience, 1:100), APC anti-mouse CD11C (Biolegend, 1:100), Brilliant Violet 510 anti-mouse CD8a (Biolegend, 1:100), and/or PerCP-Cyanine5.5 anti-mouse CD19 (eBioscience, 1:100). For P2RX1 labeling, primary antibody (Alomone, 1:80) was labeled for 30 min, and subsequently Alexa Fluor 594 anti-rabbit secondary antibody (Abcam, 1:500) were labeled for another 30 min after washing. Secondary antibody staining without primary antibody was used as negative control. A transcription factor staining set (eBioscience) was used to detect intracellular Nrf2 expression, following the manufacturer's instructions. For intracellular ROS detection, neutrophils were treated with indicated stimulus for 4 h, washed with PBS, incubated with culture medium that containing 10 μM dihydrorhodamine 123 for 30 min. Flow cytometry was performed in BD Fortessa FACS with FACSDiva software v6.0.

**Fluorescence-activated cell sorting (FACS).** Mice were euthanized and liver metastases were harvested. Tissues were carefully minced and digested with 2 mg/mL collagenase A and 1× DNase I for 30 min. Digestion was quenched by FBS and filtered with 40 μm Nylon mesh. Nonparenchymal cell were enriched by undergoing a centrifugation step through 37.5% Percoll. The cell pellets at the tube bottom were washed and resuspended in staining buffer (2%FBS in PBS).

Single cell suspensions were blocked with anti-mouse CD16/CD32 antibodies for 10 min and labeled with indicated fluorophore-conjugated antibodies at recommended dilutions and times. For P2RX1 labeling (Alomone, 1:80), primary antibody was labeled for 30 min. After washing, fluorophore-conjugated secondary antibodies were labeled for another 30 min. Before sorting, SYTOX Blue (Thermo Fisher) was added to exclude dead cells. Neutrophils were then purified with Aria-II (BD) FACS.

**Neutrophil stimulation.** Tumor conditioned medium (TCM) was prepared as previously reported[31]. Briefly, liver metastatic tissues from WT mice were plated in 1 mL RPMI 1640 containing 10% FBS for 24 h, and supernatants were harvested by centrifuging. Bone marrow-derived neutrophils were treated with 50% TCM for 12 h, with or without an anti-GM-CSF neutralizing antibody (10 μg/mL, BioLegend). In other studies, bone marrow-derived neutrophils were treated with GM-CSF (100 ng/mL, PeproTech), G-CSF (100 ng/mL, PeproTech), IFN-γ (20 ng/mL, PeproTech), LPS (100 ng/mL, Sigma), IL-4 (20 ng/mL, PeproTech) or LPS (100 ng/mL) + IFN-γ (20 ng/mL) for 12 h in RPMI 1640 containing 10% FBS and indicated tests were performed, unless otherwise stated.

**Histology and immunofluorescence staining.** Twenty cases of PDAC liver metastases and adjacent livers tissues were obtained from Ren Ji hospital from January 2015 to June 2018 (Supplementary Table 1). The study was conducted in accordance with International Ethical Guidelines for Biomedical Research Involving Human Subjects (CIOMS). The study was approved by the Research Ethics Committee of Ren Ji Hospital, School of Medicine, Shanghai Jiao Tong University. All patients had not received radiotherapy, chemotherapy, hormone therapy or other related antitumor therapies before surgery. Written informed consent was provided before enrollment. Hematoxylin and eosin (H&E) staining was performed routinely and observed with Leica Aperio ScanScope Console 12.3. For immunofluorescence staining, paraffin sections were dewaxed with gradient ethanol, steam heated for antigen retrieval in citrate-based buffer, blocked with PBS containing 10%BSA for 1 h, stained with anti-P2RX1 (Alomone, 1:200), and CD66b (Biolegend, 1:1000), Ly6G (Biolegend, 1:1000), or citrullinated histone H3 (CitH3) (Alomone, 1:200) overnight and secondary antibody (Abcam, 1:1000) for 2 hr and finally mounted in DAPI-containing media (Vector Labs) for imaging on a laser scanning confocal microscope (Leica SP8 LAS X).

**Metabolism assays.** Metabolic assays for extracellular acidification rate (ECAR) and oxygen consumption rate (OCR) were performed with the Seahorse XF96 Flux Analyser (Agilent) and Agilent Wave 2.6.1 software according to the manufacturer's instructions. Briefly, $2–4 \times 10^5$ cells were seeded on gelatin-coated plates and treated with indicated stimulus for 12 h. 1 h before testing, the media was replaced with assay media. For ECAR test, 1 μmol/L oligomycin, 0.5 μmol/L carbonyl cyanide p-trifluoromethoxyphenylhydrazone (FCCP), and 0.5 μmol/L rotenone plus 0.1 μmol/L antimycin A were injected to the wells. For OCR test, 10 mmol/L glucose, 1 mol/L oligomycin, and 50 mmol/L 2-deoxyglucose were added to the wells.

**Bone marrow transplantation.** Six to eight-weeks old CD45.1+WT and CD45.2+ mice were used for bone marrow translation studies. Recipient mice were exposed to lethally irradiation (11 Gy), and subsequently received intravenous injection of $5 \times 10^6$ donor bone marrow cells. Animals were analyzed after 6–8 weeks by flow cytometry for reconstitution of the hematopoietic system.

**Suppression assay.** CD8+ OT1 cells were isolated from spleen of OT1 mice by CD8+ magnetic bead selection (Miltenyi), and OT1 cells were labeled with 2 μmol/L carboxyfluorescein succinimidyl ester (CFSE, Thermo Fisher), according to the manufacturer's instructions. In a 3-day incubation, OT1 cells was co-cultured 1:1 with GM-CSF-primed neutrophils (100 ng/mL for 12 h) in cognate peptides SIINFEKL (10 ng/mL) and IL-2 (10 ng/mL), with or without a mouse PD-L1 neutralizing antibody (20 μg/mL, BioXCell). Proliferation of antigen activated CTLs was determined by flow cytometry. For cytotoxicity study, KPC cells were transfected with lentiviral vector (LV)-OVA or empty LV (GeneChem) at a MOI 50, and incubated with puromycin (5 μg/mL) for 5 days to select stable KPC-OVA or KPC-LV cell lines. OT1 cells were stimulated with SIINFEKL (10 ng/mL) and IL-2 (10 ng/mL) for 5 days to generate CTLs. CFSE-labeled GM-CSF-primed neutrophils and CTLs mixtures (1:1) were co-cultured with CFSE-free KPC-OVA or KPC-LV cells at a ratio of 20:1, with or without a mouse PD-1 (BioXCell, 20 μg/mL) or PD-L1 neutralizing antibody (BioXCell, 20 μg/mL) for 8 hr. Cytotoxicity was analyzed by calculating percentage of CFSE-PI+ cells using flow cytometry.

**Chromatin immunoprecipitation (ChIP) followed by PCR (ChIP-PCR).** Pierce Agarose ChiP Kit (Thermo Fisher) was applied for CHIP-PCR, following the manufacturer's instructions. Briefly, $P2rx1^{-/-}$ neutrophils were treated with GM-CSF for 12 h and formaldehyde crosslinked cells were lysed and sonicated to shear the DNA. The sonicated DNA–Protein complexes were immunoprecipitated with control IgG (CST) and Nrf2 antibodies (CST, 1:200). Primers used were showed as following: PD-L1-1-F CTTCGGTTTCACAGACAGCGG, PDL1-1-R CTGAGTTCTTAACTTTCCCAG, PD-L1-2-F GGAATAGTGATGAGGGTGTG, PDL1-2-R TTCTGAAGGGGAATGAGGAA. GAPDH primers were provided in the Kit.

**RNA-sequencing (RNA-seq).** For RNA-seq of PDAC liver metastases, total RNAs were isolated using the TRIzol reagent following the manufacturer's instructions and sent to Novogene Co., Ltd. for clustering and sequencing. For RNA-seq of FACS purified cells, reverse transcription was performed using a SMARTer Ultra Low RNA Kit (Clontech). cDNA was amplified using an Advantage 2 PCR Kit (Clontech) according to the manufacturer's protocol. Then, RNA-seq was performed by Ebioservice Co., Ltd. RNA sequencing data were collected using NextSeq System Suite v2.2.0.

**Luciferase reporter assay.** Luciferase reporter assay was performed as previously described[60]. NIH3T3 cells were transfected with pGLB3-WT-PD-L1 reporter or pGLB3-Mut-PD-L1 reporter alongside a Renilla luciferase reporter internal control. Nrf2 was overexpressed and followed by quantification of luciferase activity. Cells were lysed, and extracts were assayed in technical triplicate for firefly and Renilla luciferase activity using the Dual Luciferase Reporter Assay (Promega). WT and mutant PD-L1 plasmid were described in Supplementary Table 2.

**Immunoblotting.** Cell lysates were prepared by using RIPA lysis and extraction buffer (Sigma). After total protein normalized, protein samples were separated by 8–10% SDS-PAGE gel electrophoresis and transferred onto 0.22 μm NC membranes. After blocking with 5% skimmed milk or 3% BSA diluted in Tris buffer saline plus 0.1% Tween 20 (TBST) for 1 h at room temperature, membranes were incubated overnight with P2RX1 primary antibody (Alomone, 1:200) or β-actin (Yeasen, 1:5000), and 2 h with secondary antibody (Jackson ImmunoResearch, 1:10,000). Enhanced chemiluminescence (ECL) was performed using ECL kit (share-bio, WB012), visualized by the Bio-Rad system.

**PCR.** Total RNA extraction and RNA reversely transcription are using Trizol reagent (Takara) and PrimeScript RT-PCR kit (Takara) according to the common protocols. Real-time PCR analyses were applied for gene expression study and SYBR Premix Ex Taq (Roche) was used to run PCR on a 7500 Real-time PCR system (Applied Biosystems) at the recommended thermal settings. Relative mRNA expression was calculated using the $2^{(-\Delta\Delta Ct)}$ method and normalized to β-actin mRNA levels. Following primers were used: Nrf2-F TAAAGCTTT-CAACCCGAAGCACGC, Nrf2-R TCCATTTCCGAGTCACTGAACCCA; TNF-α-F GGAACTGGCAGAAGAGGCACTC, TNF-α-R GCAGGAATGAGAA-GAGGCTGAGAC; IL-1α-F TCTCAGATTCACAACTGTTCGTG, IL-1α-R AGAAAATGAGGTCGGTCTCACTA; P2rx1-F: GGATGGTGCTGGTAC-GAAACA, P2rx1-R: CACTGACACACTGCTGATAAGG; P2ry2-F: CTGGTGCGTTTCCTCTTCTAC, P2ry2-R: CGCTGGTGGTGACGAAGTA; P2ry10-F: TTTCCATCCTTTTTCCCTATGCC, P2ry10-R: AGTTGGTCAC-GAAACTCTGAAG; β-actin-F ATGGATGACGATATCGCT, β-actin-R ATGAGGTAGTCTGTCAGGT.

Purification of DNA was performed using DNeasy kits (Qiagen). To quantify the mitochondrial DNA (mtDNA)/nuclear DNA (nDNA) ratio, qPCR was used to amplify one gene from the mitochondrial genome (mtATP6) and one gene from the nuclear genome (Rpl13a) as previously reported[61]. Primer sequences were as follows: mtATP6-F CAGTCCCCTCCCTAGGACTT, mtATP6-R TCAGAGCATTGGCCATAGAA; Rpl13a-F GGGCAGGTTCTGGTATTGGAT, Rpl13a-R GGCTCGGAAATGGTAGGGG.

**Bioinformatics analysis**. Gene Expression Omnibus (GEO) dataset GSE71729 containing 145 primary PDAC, 46 adjacent pancreases, 25 liver metastases and 27 adjacent livers were used to investigate the differential transcriptomic signatures. The microarray data were normalized and log2- transformed, and then differential analyses were performed using the R package the Linear Models for Microarray Data (Limma). The immunome of immune cell types infiltrating tumors was conducted as previously reported[14]. Single cell sequencing of pancreas[62] and liver tissues[63] was analyzed with Bioturing software. Distribution of P2RX1 in different tissues and cell lines was screened using Human Protein Atlas database available from http://www.proteinatlas.org. Gens set enrichment analysis (GSEA) was performed on the Broad Institute Platform and statistical significance (false discovery rate, FDR) was set at 0.25. Nrf2 target genes were obtained from chromatin enrichment analysis (ChEA) transcription factor binding site profiles NRF2-20460467-MEF-MOUSE[37].

**Study approval**. Clinical samples studies were approved by the Research Ethics Committee of Ren Ji Hospital, School of Medicine, Shanghai Jiao Tong University. Animal experiments were approved by Institutional Animal Care and Use Committee of East China Normal University.

**Statistics**. Data are presented as mean ± standard deviation. Reproducibility was ensured by performing more than three independent experiments. All statistics were carried out using GraphPad Prism 7.0.4. Student's $t$ test was used to compare the differences between two groups. One-way analysis of variance (ANOVA) and Tukey's test were used for the three or more groups comparisons. Survival was analyzed with log-rank test. A value of $P < 0.05$ was considered to be statistically significant.

**Reporting summary**. Further information on research design is available in the Nature Research Reporting Summary linked to this article.

## Data availability

The PDAC liver metastases transcriptome data used in this study are available in the GEO database under accession code GSE71729 and GSE151580. The RNA-seq data generated in this study have been deposited in the Sequence Read Archive (SRA) repository under accession code PRJNA664673. The remaining data are available within the Article, Supplementary Information, or available from the authors upon request. Source data are provided with this paper.

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

## Acknowledgements

We thank Prof. Jing Xue for the gift of KPC mice-derived PDAC cell line FC1199; Prof. Keli Yang for the gift of OT1 mice. This study was supported by the National Natural Science Foundation of China (ID 81672358 to Z.-G. Zhang; ID 81701945 to X. Wang; ID 81802890 to X.-L. Zhang; ID 81871923 to J. Li; ID 81902370 to S.-H. Jiang; ID 81874175 to Y.-W. Sun), Program of Shanghai Academic/Technology Research Leader (ID 19XD1403400, to Z.-G. Zhang), Shanghai International Science and Technology Cooperation Fund (ID 18410721000, to Z.-G. Zhang) and Excellent Academic Leader of Shanghai Municipal Health Bureau (ID 2018BR32, to Z.-G. Zhang), the Natural Science Foundation of Shanghai (ID 18ZR1436900 to X.-L. Zhang), the China Postdoctoral Science Foundation (ID 2018M640403 to X. Wang), Innovative Research Team of High-Level Local Universities in Shanghai (to Y.-W. Sun), Shanghai Science and Technology Committee (ID 17411952100 to Y.-W. Sun), Medical Transformation Crossing Funding from Shanghai Jiao Tong University (ID ZH2018ZDB08 to Y.-W. Sun).

## Author contributions

Z.-G. Zhang conceived the project. X.-L. Zhang, S.-H. Jiang, Y.-W. Sun, and D.-Y. Chen designed experiments, and interpreted data in the manuscript. X. Wang, L.-P. Hu, W.-T. Qin, Q. Li, K.-X. Zhou, P.-Q. Huang, C.-J. Xu, G.-A. Tian, J.-Y. Yang, M.-W. Yang, D.-J. Liu, L.-L. Yao, Y.-H. Wang, J. Li performed the experiments. Q. Yang performed bioinformatics analyses. The contributions of X.W., L.-P. Hu, W.-T. Qin, Q. Yang, and D.-Y. Chen were equal in significance to the final conclusions of the study; therefore, the authorship order reflects their temporal involvement in the project.

## Competing interests

The authors declare no competing interests.
