## [Peer Review File · Nature Communications]

REVIEWER COMMENTS

Reviewer #1 (Remarks to the Author):

This is an interesting and timely study by a group that has made important contributions to this field in the recent past. Main findings are that 1) P2X1-neg neutrophils accumulate at sites of PDAC metastasis, 2) P2X1-neg neutrophils express high levels of immunosuppressive molecules and show an enhanced mitochondrial metabolism, 3) Nrf2 is upregulated in P2X1-neg neutrophils, 4) immunosuppression mediated by P2X1-neg neutrophils seems to be dependent on PD-1 expression. To my knowledge, this is the first solid demonstration that the P2X1 receptor is involved in the modulation of immunosuppressive (or immunostimulatory) functions in the tumor microenvironment as well as in other settings. This study explores in depth the underlying mechanism with a wealth of sophisticated techniques. However, a few issues need further attention.

Major criticisms

1. The study by Zhang and co-workers shows that P2X1-neg neutrophils have a more active oxidative metabolism. Previous studies in human lymphocytes and human leukemic cell lines suggested that P2X1 activity is necessary for a "healthy" mitochondrial metabolism (PMID: 27020575; PMID: 26150546). How do the Authors reconcile their data with this previous evidence?
2. Evidence obtained using the P2X1-KO model is very convincing, however from the stand point of translational medicine it will be more convincing to show that administration of P2X1 blockers to WT mice with liver metastasis has a similar effect. Some P2X1 antagonists are available from Tocris.
3. I noticed that Th17 cells are also upregulated in liver metastasis (Fig. 1B). Could the Authors comment on this?
4. I am confused by Fig. S2A, which shows that P2X1 positively correlated with infiltration of antitumor immune cells, notably CD8 central memory, type 1 helper cells and others. Yet, the Authors also report (lines 164-165) that the P2X1 receptor was barely expressed by CD4+, CD8+ and CD19+ lymphoid cells. Sure, P2X1 expression by lymphoid cells, especially mouse lymphoid cells, is not very well known, however I am somewhat surprised by these findings. The Authors may wish to check previous papers (PMID 26150546; PMID 206602288; PMID 26853442).
5. The Authors state that normal pancreatic tissue does not express P2X1, however this seems not to be the case looking at Fig. S3B.
6. In Fig. 5E it is shown that N1- or N2-polarized neutrophils have an enhanced mitochondrial mass. However, N1-polarized neutrophils have reduced OxPhos activity, thus I would anticipate a lower mitochondrial mass compared to N2-polarized neutrophils.

Reviewer #2 (Remarks to the Author):

Overall this paper presents very interesting data showing that P2RX1 may distinguish 2 subsets of neutrophils with differing functions. After demonstrating the presence of these two the authors explore the functional differences between these two neutrophil populations using a P2RX1-/- mouse. The manuscript makes less clear that signalling through P2RX1 leads to the alterations in the neutrophil function. Either way, as cause or effect this could be very interesting. However there are several serious concerns with the data as presented

Specific comments:

In general in the cell culture experiments the n for each group is shown but the number of independent experiments performed is not.

Fig 1 shows a transcriptomic analysis of a publicly available data set of gene expression from pancreatic adenocarcinomas, adjacent pancreas, adjacent liver and liver metastases. Analysis of the immune cell gene expression from these data showed lesser levels of CD8 T cells among many differences in liver metastases compared to the other tissue types examined. Further GO Biological Analysis found decreased purinergic nucleotide receptor signalling and the authors focused on this finding also finding decreased expression of the purinergic receptor P2RX1 in liver metastases compared to the other tissues. Additionally a variety of the RNAs for other molecules involved in purinergic signalling were also found to be altered with P2RX1 downregulated in both liver metastases compared to primary cancers as well as compared to adjacent liver. Hence they focussed upon this receptor.

I wonder at this point whether they could also go back to Fig. 1B and try to identify neutrophils based on gene expression profiles as neutrophils are so important for the rest of the paper.

At this point they generated a mouse genetically deficient in P2RX1

Fig. 1D units on Y axis not clear.

They then generated a P2RX1 knockout mouse. No details about the health or properties of this mouse other than its genotype are shown. For example what are its neutrophil levels in the blood, T cells etc, weight? What about cellular composition of the bone marrow?

After intrasplenic injection (of a very large number of cells) they saw greater growth of liver metastases based on bioluminescence in the P2RX1^{-/-} mice. However the difference in luminescence- at least 10 fold- was not clearly 10 fold in the volume of metastases at 17 days in Fig 2E. Do they have any other parameters of the extent of liver metastases in the mice harvested at 17 days? Ki67 staining is shown at 17days but no quantitative analysis is provided. In this image glandular structures are apparent in the metastasis where WT is the host but not in P2RX1^{-/-}. Is this a consistent finding?

A variety of immune checkpoint molecules are differentially expressed between the liver metastases in the different hosts.

Figure 3. Splenic cells from WT mice were evaluated for P2RX1 expression. The Y axis in A and the dotted lines should be better labelled and explained. The F4/80 channel cannot be seen. In B the gating strategy is not presented. In reality there are too few cells in the liver metastases to make this comparison. In C and D they investigate immunohistochemistry on liver mets from KPC tumors and from 20 patients. No details at all are provided about the patient specimens. This information needs to be provided.

Here they show that the percentage of neutrophils that are P2RX1⁺ in liver metastases is less than in adjacent liver. Although p values are shown the variances are not indicated.

They then create bone marrow chimeras with WT and P2RX1^{-/-} cells to ask whether the difference in liver metastatic growth is due to bone marrow derived cells or to other host cells and the answer is that it is bone marrow derived cells. They also ask whether elimination of neutrophils by Ly6G administration alters the effect and see that anti Ly6G greatly reduces metastasis. There is a lesser point in that they do not show the extent to which Ly6G has indeed eliminated neutrophils and this should be shown somewhere. More important however is that neutrophils have been

shown to enhance liver metastasis by generation of neutrophil traps (see- Neutrophil extracellular traps promote liver micrometastasis in pancreatic ductal adenocarcinoma via the activation of cancer associated fibroblasts.

Takesue S, et al *Int J Oncol.* 2020 Feb;56(2):596-605. doi: 10.3892/ijo.2019.4951. Epub 2019 Dec 24. And Primary tumors induce neutrophil extracellular traps with targetable metastasis promoting effects. Rayes RF, et al. *JCI Insight.* 2019 Jul 25;5. pii: 128008. doi: 10.1172/jci.insight.128008.)

(NETS) so that removal of neutrophils at the time of intrasplenic injection will greatly limit seeding of the liver and so reduce metastasis. In fact their data that anti-ly6g reduces metastasis to barely detectable levels in both host and mutant genotypes is entirely consistent with this possibility. This will be independent of any form of adaptive immunity as it would occur in the first 24 hours after injection. To determine whether P2RX1 affects colony growth after seeding, Ly6G depletion at say 3-4 days after injection should be tested for its effect on colony growth. The differential may then be evident.

Fig 4 examines the numbers of Ly6G+ cells in the BM and the blood in naive mice compared to mice bearing liver metastases with increased positive cells in the BM and in the blood. The level of cells 8.5% in the blood seems a bit low for mice. They treat isolated BM neutrophils with various cytokines and examine gene expression seeing a decrease in expression. This was only performed once so that statistical significance is not available. Because the neutrophils in WT mice that are P2RX1^{-/-} may be different from P2RX1^{-/-} neutrophils, some demonstration that the differences are consistent might be helpful while understanding that obtaining sufficient material might be difficult. IHC might help.

Fig. 5 neutrophils were isolated from liver metastases in WT mice and sorted to generate a population of P2RX1⁺ neutrophils to compare to P2RX1⁻ neutrophils isolated from liver metastases in P2RX1^{-/-} mice in gene expression finding enhanced glycolytic pathway genes in P2RX1⁺. They then examined ECAR and OCR in these neutrophil populations seeing enhanced OCR and decreased ECAR in neutrophils from the P2RX1^{-/-} liver metastases after stimulation with LPS or IL-4, results consistent with the gene expression differences. They also determine mitochondrial mass using mitotracker green. Because mitotracker green measures redox this is only a reflection of mitochondrial mass in the two populations at the same redox state. In this case where OCR differs so the redox states will differ, it would be more appropriate to use a marker of mitochondrial mass, such as determination of a mitochondrial protein or amount of DNA etc to verify the conclusion about difference in mass.

In Fig 6 they now ask whether the difference in Nrf pathways found in the gene expression analysis might be directed by P2RX1 signaling. Fig. S7 showed that Nrf2 RNA was greater in P2RX1^{-/-} neutrophils than P2RX1⁺ tumor derived neutrophils and that stimulation with a variety of cytokines led to increased Nrf2 to a greater extent in P2RX1⁻ than P2RX1⁺ neutrophils. In fact a Nrf2 inhibitor reduced the response of increased OCR or decreased ECAR to LPS+IFN γ or IL-4. IL-1 and TNF expression were affected.

They now ask about differences in PD-L1 expression on the different neutrophil populations. Fig 7 shows that a substantial proportion of CD8 cells from P2RX1^{-/-} mice in the liver metastases express PD-L1 more than from WT liver metastases. They do not show the expression on the KPC cells, which have been found to express PD-L1 in other settings. Neutrophils from P2RX1^{-/-} liver metastases also expressed more PD-L1 than those from from P2RX1^{+/+} mice. They then make the case that Nrf2 controls PD-L1 expression in neutrophils. Inhibition of Nrf2 blocks induction of PD-L1 by GM-CSF or tumor conditioned medium. They show that making a prediction of the PD-L1 promoter based on integrative genomics that binding to the predicted regions can be found for Nrf2. Here one might ask for more demonstration that the presumed regions are actually part of the promoter activity before coming to the conclusion that Nrf2 directly regulated PD-L1 transcription.

They also use activated T cells to show that addition of isolated neutrophils can reduce their proliferation with a reduction from 42% to 25% by the WT but 19% by the P2RX1^{-/-} neutrophils. This seems unlikely to be significant. Anti PD-L1 also led to a reduction. However they do not indicate whether their T cells express PD-L1. A control with added anti PD-L1 is not shown. The same problem of potential T cell PD-L1 holds in the cytotoxicity assay as well as PD-L1 on the cancer cells.

Minor comments:

The sentence line 31 "Mechanistically, the transcription factor Nrf2 was upregulated in P2RX1-deficient neutrophils and accounted for PD-L1 expression and metabolic reprogramming." I feel is an overstatement. I think they have demonstrated a correlation but not causation.

Line 78 what is a "complete" microenvironment?

Line 111 The references for the statement "The roles of ADORA2B in promoting PDAC progression 111 and tumor metastasis have been well characterized" but 14, 15 don't really support this statement as regards pancreatic cancer.

Misspelling pancreas in SF1

Fig 6 Cannot really see blue filled vs black

Line 336 "These results suggest that increased intracellular ROS in P2rx1^{-/-} neutrophils contributes to PD-L1 expression by activating the ROS-sensitive transcription factor Nrf2." This seems to overstate the correlation described

Line -362 "By upregulating the expression of PD-L1 in a ROS/Nrf2dependent manner, P2RX1-neutrophils exhibited potent immunosuppressive effects on PDAC-specific CTLs." Again overstates the correlation shown.

Could have been more in depth exploration of the connection between P2RX1 and NRF

Line 376 "Considering that glycolysis frequently supports immunostimulatory responses, whereas mitochondrial OXPHOS frequently supports immunosuppressive responses, we suggest that Nrf2 supports the plasticity of the immunosuppressive phenotype of P2RX1-378 deficient neutrophils." I don't understand what this means?

Line 389 "In the present study, we identified that liver metastatic PDAC evades antitumor immunity in the presence of accumulated extracellular ATP by mobilizing and recruiting P2RX1-neutrophils." Don't actually show this- it's a hypothesis for the discussion.

Line 392 "As heterogeneous expression of purinergic receptors is also reported in other immune cell types, this strategy is likely common for other pathological conditions." What does this mean?

Reviewer #3 (Remarks to the Author):

In this paper the authors described a role for P2RX1-negative neutrophils in pancreatic cancer liver metastases. Starting from bioinformatic analysis of available data base, they showed an association between P2RX1 expression and immunosuppressive microenvironment in PDAC liver metastasis. Authors confirmed these findings by performing in vivo experiments on P2RX1 knockout mice, in which they showed that the absence of P2RX1 led to an elevated level of PDAC liver metastasis. Flow cytometry analysis demonstrated that among all leukocyte infiltrating tumor microenvironment, the expression of P2RX1 is found higher in neutrophils. They continued with RNA seq analysis and characterized P2RX1-negative neutrophils. The authors suggested that the transcription factor Nrf2 is crucial in the immunosuppressive role of P2RX1-negative neutrophils by upregulating PD-L1 expression.

The manuscript is well presented, and the figures are organized well enough to catch the different points quickly.

Overall, the paper is interesting but some revisions are needed.

1: Considering P2RX1 as the key element in this manuscript, there is a need for a short description on purinergic receptors and their role in cancer or leukocyte trafficking in the introduction.

2: A recent paper (Patel, S et al. Nat Immunology 2019) showed that autocrine ATP signaling in neutrophils through purinergic receptors shape their activation state. Why this paper is not discussed?

3: Line 84-85: The authors claimed to analyze the immunome and quoted the reference 11 (Charoentong P. et al). Charoentong P. et al. announced 28 leukocytes while in this manuscript, 26 of them have been analyzed. More importantly in this analysis neutrophils are lacking, as neutrophils are the center of this manuscript they should be included in this analysis.

4: Even though later the authors described P2RX1 to be highly expressed in neutrophils, in their bioinformatic analysis in which they showed P2RX1 was positively correlated with antitumor immunocyte infiltration, they do not show the analyses for neutrophils (Figure S2A). Again, as neutrophils are important in the study, they should be inserted. Perhaps neutrophils are "hidden" with MDSC?

5: In the Figure 5 or S6B, what is the relevance to stimulate neutrophils with IFN γ , IL4 and LPS for your model? In particular for LPS?. These in vitro experiments should be performed with tumor conditioned medium as performed in other parts of the paper.

6: Authors should insert references for the N1 and N2 markers they have chosen. Recent evidence has shown that MET signaling in neutrophils can induce also an immunosuppressive phenotype (Glodde, N. et al. Immunity 2017).

7: The phenotype of CXCR4+ P2RX1-negative neutrophils could be explored more deeply. Are they immature neutrophils, aged neutrophils, or mature activated neutrophils?

Minor comments

1. Along the text whenever some results are described, the corresponding figure, or a reference, should be mentioned, here are some examples in which in this manuscript this aspect is ignored:

- Page5, line 90

- Page7, line 151

- Page10, line 201 (Figure 4 is not describing the mentioned information)

2. In figure 3A the results are shown only by histograms, but as in the text the frequency of cells expressing P2RX1 is also mentioned, it would be useful to indicate the percentage of positive cell in the histograms.

Reviewer #1 (Remarks to the Author):

This is an interesting and timely study by a group that has made important contributions to this field in the recent past. Main findings are that 1) P2X1-neg neutrophils accumulate at sites of PDAC metastasis, 2) P2X1-neg neutrophils express high levels of immunosuppressive molecules and show an enhanced mitochondrial metabolism, 3) Nrf2 is upregulated in P2X1-neg neutrophils, 4) immunosuppression mediated by P2X1-neg neutrophils seems to be dependent on PD-1 expression. To my knowledge, this is the first solid demonstration that the P2X1 receptor is involved in the modulation of immunosuppressive (or immunostimulatory) functions in the tumor microenvironment as well as in other settings. This study explores in depth the underlying mechanism with a wealth of sophisticated techniques. However, a few issues need further attention.

Major criticisms

1. The study by Zhang and co-workers shows that P2X1-neg neutrophils have a more active oxidative metabolism. Previous studies in human lymphocytes and human leukemic cell lines suggested that P2X1 activity is necessary for a "healthy" mitochondrial metabolism (PMID: 27020575; PMID: 26150546). How do the Authors reconcile their data with this previous evidence?

Response: We appreciate the reviewer for bringing these highly relevant references to our attention.

Wolfgang team revealed that autocrine-induced P2RX1 activation was essential for human CD4+ T cells and Jurkat cells activation and proliferation^{1,2}. Using P2RX1 antagonist NF023 or NF157 to treat naïve or activated CD4+ T cells for 20 min, they found that mitochondrial activity was reduced¹. In addition, treating Jurkat cells with NF023 or NF157 for 10 min, they obtained similar results². The mechanism was likely to be associated with P2RX1-involved cytosolic Ca²⁺ response, and mitochondrial Ca²⁺ uptake.

In line with these, our previous study also observed that autocrine-induced P2RX1 activation promotes neutrophil Ca²⁺ influx, and cytosolic Ca²⁺ subsequently reinforces

neutrophils' responses to bacterial LPS within 10 min³. It is noticed that under physiological conditions, the rapid accumulation of cytosolic Ca²⁺ stimulates mitochondrial oxidative metabolism through the modulation of Ca²⁺-sensitive dehydrogenases and metabolite carriers⁴. But, a long term of cytosolic Ca²⁺ overload is harmful to mitochondrial activity directly⁵, and dysregulated Ca²⁺ influx may reduce mitochondrial activity indirectly by inhibiting Nrf2⁶. We suppose that the autocrine-released ATP from CD4+ T cells (or leukemic cell) and neutrophils is essential to support the normal activation state. However, in the ATP-rich tumor microenvironment, exogenous ATP-induced sustained P2RX1 activation and the subsequent long-term Ca²⁺ influx may be detrimental for P2RX1+ neutrophil mitochondrial metabolism.

This issue has been discussed in the Discussion section of revised manuscript.

2. Evidence obtained using the P2X1-KO model is very convincing, however from the stand point of translational medicine it will be more convincing to show that administration of P2X1 blockers to WT mice with liver metastasis has a similar effect. Some P2X1 antagonists are available from Tocris.

Response: We thank the reviewer for the helpful suggestion. A highly selective P2RX1 antagonist, NF449, was obtained from Tocris and 10 mg/kg dosage of NF449 was i.p. administrated daily. At days 17, we observed that P2RX1 antagonist NF449 significantly promoted metastatic tumor growth in livers. This result is similar with the previous results obtained using P2RX1-KO mice, and has been added in the revised manuscript (Figure S4A).

3. I noticed that Th17 cells are also upregulated in liver metastasis (Fig. 1B). Could the Authors comment on this?

Response: We thank the reviewer for this valuable suggestion. Comments on Th17 in PDAC tumor microenvironment has been updated in the Results section of the revised manuscript as following:

“Apart from Th1 and Th2 cells, the role of Th17 cells in tumor microenvironment has recently attracted much attention, with both promotive¹⁶ and suppressive¹⁷ effects on tumor metastasis having been reported. Our analyses showed that primary PDAC and PDAC liver metastases had more Th17 cells as compared to adjacent pancreas and adjacent liver tissues, respectively (Figure 1B). The specific function of Th17 in PDAC progression remains to be explored.”

4. I am confused by Fig. S2A, which shows that P2X1 positively correlated with infiltration of antitumor immune cells, notably CD8 central memory, type 1 helper cells and others. Yet, the Authors also report (lines 164-165) that the P2X1 receptor was barely expressed by CD4+, CD8+ and CD19+ lymphoid cells. Sure, P2X1 expression by lymphoid cells, especially mouse lymphoid cells, is not very well known, however I am somewhat surprised by these findings. The Authors may wish to check previous papers (PMID 26150546; PMID 206602288; PMID 26853442).

Response: We appreciate the reviewer’s important comments.

To identify specific roles of purinergic signaling in the immune microenvironment of PDAC liver metastases, we analyzed the correlations between purinergic signaling molecules expression and PDAC liver metastatic immunocyte components. The heatmap results in Fig. S2A showed that CD8 central memory, type 1 helper cells and other antitumor immune cells are positively correlated with P2RX1 expression. We propose that this observation can be explained by: 1) P2RX1 may be directly expressed in these antitumor immunocytes; 2) P2RX1 expressed in other cell types may affect the accumulation of antitumor immunocytes in PDAC liver metastasis.

As the reviewer points out, three well-executed papers showed that P2RX1 is important for the activation of lymphocytes. We noticed that in the paper PMID 20660288, the authors found that P2RX1 mRNA expression was detected in primary CD4+ T cells (PMID 20660288, Fig. 1A). However, compared to the highly expressed P2RX4 and P2RX5, P2RX1 mRNA expression level was hundreds of times lower (PMID 20660288, Fig. 1A). Interestingly, when activated by PMA+PHA for 4 h *in vitro*, P2RX1 mRNA expression was upregulated by about 1500 times (PMID

20660288, Fig. 1C). It indicates that P2RX1 expression on CD4+ T cells depends on the activation states. In two independent public human immunocyte RNA sequencing database HPA and Monaco, we observe that P2RX1 expression is barely detected in naïve and memory lymphocytes (see figures below). Instead, in myeloid lineage cells, including neutrophils, monocytes and DCs, P2RX1 is highly expressed (see figures below). Therefore, we suggest that in the metastatic microenvironment, P2RX1 expression is not likely to be contributed by lymphocytes.

To further address this concern, we have improved our flow cytometry experiments by enriching immunocytes with Percoll gradient centrifugation. In addition, P2RX1 was simultaneously stained with immune cell markers CD45, Ly6G, F4/80, CD11c, CD4, CD8 and CD19, and then detected in one flow cytometry panel. New flow cytometry studies were consistent with our previous results, indicating that P2RX1 was lowly expressed in CD4+, CD8+ and CD19+ lymphoid lineage cells (Figure 3C and S5D-E). In addition, P2RX1, which highly expressed in Ly6G+ neutrophils, was markedly reduced in PDAC liver metastases. These *in vivo* investigations further

verify our study on the expression pattern of P2RX1 in murine PDAC liver metastases.

5. The Authors state that normal pancreatic tissue does not express P2X1, however this seems not to be the case looking at Fig. S3B.

Response: We appreciate the reviewer's helpful comment.

In the manuscript, we focused on the P2RX1 expression in PDAC liver metastasis. Therefore, we only mentioned the expression of P2RX1 in hepatic cells and pancreatic ductal cells based on the public single-cell RNA-seq database, and which was described as following:

“The public single-cell RNA-seq database revealed that P2RX1 was barely detected in **hepatic cells** or **pancreatic ductal cells** (Figure S3A).”

Interestingly, when we revisited the P2RX1 expression in pancreas single-cell RNA-seq database, as the reviewer pointed, we noticed that P2RX1 was highly expressed in a portion of pancreas acinar cells (Figure S3A). We thank the reviewer's reminder. To avoid further misunderstanding, we have revised this sentence as following:

“The public single-cell RNA-seq database revealed that P2RX1 was barely detected in hepatic cells of liver tissue, or pancreatic ductal cells of pancreas tissue (Figure S3A).”

6. In Fig. 5E it is shown that N1- or N2-polarized neutrophils have an enhanced mitochondrial mass. However, N1-polarized neutrophils have reduced OxPhos activity, thus I would anticipate a lower mitochondrial mass compared to N2-polarized neutrophils.

Response: We thank the reviewer for bringing this concern. As the reviewer pointed, in original Fig. 5E and Fig. S6C, we compared neutrophil mitochondrial mass in N1- or N2-polarized neutrophils by staining MitoTracker Green. Results showed that higher mitochondrial mass was observed in N1- and N2-polarized *P2rx1*^{-/-} neutrophils as compared to WT neutrophils. The reviewer further pointed that N1-polarized

neutrophils had reduced OXPHOS activity, and anticipated a lower mitochondrial mass compared to N2-polarized neutrophils. Indeed, we detected OXPHOS in N2-polarized neutrophils but not N1-polarized neutrophils (Original Fig. 5D, Revised Fig. S7F). Therefore, we speculate that the reviewer may have concerns about the relationship between MitoTracker Green staining and OXPHOS activity, which is similar to the concern of Reviewer #2. Reviewer #2 pointed that MitoTracker Green measures redox and reflects mitochondrial mass in the two populations at the same redox state, and indicates that detection of mitochondrial protein or DNA is required to verify the conclusion of mitochondrial mass.

Therefore, we further examined the mitochondrial DNA (mtDNA)/nuclear DNA (nDNA) ratio in WT and *P2rx1*^{-/-} neutrophils. Our results showed that in N1- and N2-polarized neutrophils, mtDNA/nDNA ratio was upregulated in *P2rx1*^{-/-} neutrophils as compared to WT neutrophils (Figure S7H). These data were consistent with the results of our MitoTracker Green (Figure S7G) staining studies, and suggested an enhanced mitochondrial mass in polarized *P2rx1*^{-/-} neutrophils.

Reviewer #2 (Remarks to the Author):

Overall this paper presents very interesting data showing that P2RX1 may distinguish 2 subsets of neutrophils with differing functions. After demonstrating the presence of these two the authors explore the functional differences between these two neutrophil populations using a P2RX1^{-/-} mouse. The manuscript makes less clear that signalling through P2RX1 leads to the alterations in the neutrophil function. Either way, as cause or effect this could be very interesting. However there are several serious concerns with the data as presented

Specific comments:

In general in the cell culture experiments the n for each group is shown but the number of independent experiments performed is not.

Response: We appreciate the reviewer's helpful comment. We have updated the number of independent experiments in the figure legends of the revised manuscript.

Fig 1 shows a transcriptomic analysis of a publicly available data set of gene expression from pancreatic adenocarcinomas, adjacent pancreas, adjacent liver and liver metastases. Analysis of the immune cell gene expression from these data showed lesser levels of CD8 T cells among many differences in liver metastases compared to the other tissue types examined. Further GO Biological Analysis found decreased purinergic nucleotide receptor signalling and the authors focused on this finding also finding decreased expression of the purinergic receptor P2RX1 in liver metastases compared to the other tissues. Additionally, a variety of the RNAs for other molecules involved in purinergic signalling were also found to be altered with P2RX1 downregulated in both liver metastases compared to primary cancers as well as compared to adjacent liver. Hence they focussed upon this receptor.

I wonder at this point whether they could also go back to Fig. 1B and try to identify neutrophils based on gene expression profiles as neutrophils are so important for the rest of the paper.

Response: We appreciate the reviewer for noticing that this important information was missing. Neutrophils and eosinophils have now been included in revised Fig. 1B and Fig. S2A. In addition, we have added comments on this results in the Results section of revised manuscript:

“Emerging evidence suggests that neutrophils play important roles in tumor metastasis⁷. We found that neutrophils were rare in primary PDAC and adjacent pancreas (Figure 1B). However, in PDAC liver metastases and adjacent liver tissues, neutrophils were among the most abundant immunocytes, which suggested that they might have potential roles in PDAC liver metastasis.”

At this point they generated a mouse genetically deficient in P2RX1

Fig. 1D units on Y axis not clear.

Response: Thanks for your kind reminder. We have updated units of Y axis in Fig. 1D.

They then generated a P2RX1 knockout mouse. No details about the health or properties of this mouse other than its genotype are shown. For example what are its neutrophil levels in the blood, T cells etc, weight? What about cellular composition of the bone marrow?

Response: We thank the reviewer for this helpful comment. Details about the health properties of P2RX1 knockout mice have been provided in the Methods section of the revised manuscript:

“No obvious differences of selective fertilization/mortality in utero, development, and growth were observed between *P2rx1*^{-/-} and WT littermates. We found that genetic ablation of P2RX1 resulted in male infertility, which was consistent with a previous report indicating that P2RX1 was involved in neurogenic vas deferens contraction⁸.”

In addition, with regard to the differences of immune cells composition in WT and *P2rx1*^{-/-} mice, flow cytometry studies were performed to detect CD45+, Ly6G+, F4/80+, CD11c+, CD4+, CD8+ and CD19+ cells in spleen, bone marrow and blood.

We observed no significant differences between $P2rx1^{-/-}$ mice and WT littermates. New data have been provided in Figure S5A in the revised manuscript.

After intrasplenic injection (of a very large number of cells) they saw greater growth of liver metastases based on bioluminescence in the $P2RX1^{-/-}$ mice. However the difference in luminescence- at least 10 fold- was not clearly 10 fold in the volume of metastases at 17 days in Fig 2E. Do they have any other parameters of the extent of liver metastases in the mice harvested at 17 days?

Response: We thank the reviewer for raising this important issue. In the present study, firefly luciferase transfected KPC cells were intraperitoneally injected to seed livers of WT and $P2rx1^{-/-}$ mice, and metastatic burden was quantified by measuring luciferin emission *in vivo*. As the reviewer pointed, the difference in luminescence- at least 10 fold- was not clearly 10 fold in the volume of metastases at 17 days in Fig 2E.

We propose that the discrepancy is caused by that the bioluminescence is based on luciferase-based enzymatic reaction, and the luciferin emission is not strictly linearly correlated with the cell density. An *in vitro* experiment was conducted to measure the luciferin emission of different density of KPC cells. We observed that the 2-fold linear increase in KPC cell density did not achieve a similar linear increase in luciferin emission (See below). The increase of luciferin emission was sharper than cell density.

Following the reviewer's suggestion, we added another parameter, the liver weight, to further assess the metastatic burden as previously reported^{9, 10}. In line with the bioluminescence results, we found that liver weight was significantly higher in

P2rx1^{-/-} mice as compared to WT mice. New results have been added in the revised manuscript (Figure 2F and S4A).

Ki67 staining is shown at 17 days but no quantitative analysis is provided. In this image glandular structures are apparent in the metastasis where WT is the host but not in *P2RX1*^{-/-}. Is this a consistent finding?

Response: We appreciate the reviewer for raising this important question.

In the immunohistochemical staining images of the Ki67, we can observe the abundant distribution of glandular structures in both WT and *P2rx1*^{-/-} mice (see below). Due to that the tumor growth was more vagarious in *P2rx1*^{-/-} mice, the glandular structures were likely to be squeezed in *P2rx1*^{-/-} mice. To avoid the potential confusions, we have now updated the Ki67 staining results by presenting a more representing area. In addition, following the reviewer's suggestion, quantitative analysis of Ki67 staining has been added in Figure S4D in the revised manuscript.

A variety of immune checkpoint molecules are differentially expressed between the liver metastases in the different hosts.

Figure 3. Splenic cells from WT mice were evaluated for P2RX1 expression. The Y axis in A and the dotted lines should be better labelled and explained. The F4/80 channel cannot be seen.

Response: We thank the reviewer for the helpful comments. The Y axis has been revised in the figure; the dotted lines have been explained in the figure legend; the color of F4/80 channel has been updated.

In B the gating strategy is not presented. In reality there are too few cells in the liver metastases to make this comparison.

Response: We appreciate the reviewer for this important comment. To examine the P2RX1 expression pattern in immune cells, single cell suspension obtained from bulk metastatic or adjacent liver tissues were stained for flow cytometry analyses. As the reviewer pointed, immune cells consist a relatively small proportion of the liver metastases, flow cytometry analyzing bulk tissue suspension is less efficient. To address reviewers' concern, we have improved our flow cytometry experiments by enriching immunocytes from bulk tissues with Percoll gradient centrifugation.

In addition, P2RX1 was simultaneously stained with immune cell markers CD45, Ly6G, F4/80, CD11c, CD4, CD8 and CD19, and then detected in one flow cytometry panel. We found that new results were similar to the previous results, indicating that P2RX1 is significantly downregulated in Ly6G⁺ neutrophils. The methods, gating strategies (Figure S5D) and results (Figure 3C and S5E) have been provided in the revised manuscript.

In C and D they investigate immunohistochemistry on liver mets from KPC tumors and from 20 patients. No details at all are provided about the patient specimens. This information needs to be provided. Here they show that the percentage of neutrophils that are P2RX1⁺ in liver metastases is less than in adjacent liver. Although p values are shown the variances are not indicated.

Response: We thank the reviewer for these important comments. The patient information has been now provided in Supplementary Table 1. The percentage of P2RX1-CD66b neutrophils has now been presented with dot plots in Fig. 3F.

They then create bone marrow chimeras with WT and P2RX1^{-/-} cells to ask whether the difference in liver metastatic growth is due to bone marrow derived cells or to other host cells and the answer is that it is bone marrow derived cells. They also ask whether elimination of neutrophils by Ly6G administration alters the effect and see that anti Ly6G greatly reduces metastasis. There is a lesser point in that they do not show the extent to which Ly6G has indeed eliminated neutrophils and this should be shown somewhere.

Response: Thanks for the reviewer's reminding. In the present study, 12.5 μ g of anti-Ly6G antibodies were intraperitoneally administered daily. More than 90% neutrophils in blood and liver could be removed at 24 h after injection of 12.5 μ g of anti-Ly6G antibody (see below). This information has now been provided in the Methods section of the revised manuscript.

More important however is that neutrophils have been shown to enhance liver metastasis by generation of neutrophil traps (see- Neutrophil extracellular traps promote liver micrometastasis in pancreatic ductal adenocarcinoma via the activation of cancer associated fibroblasts.

Takesue S, et al Int J Oncol. 2020 Feb;56(2):596-605. doi: 10.3892/ijo.2019.4951. Epub 2019 Dec 24. And Primary tumors induce neutrophil extracellular traps with targetable metastasis promoting effects. Rayes RF, et al. JCI Insight. 2019 Jul 25;5. pii: 128008. doi: 10.1172/jci.insight.128008.) (NETS) so that removal of neutrophils at the time of intrasplenic injection will greatly limit seeding of the liver and so reduce metastasis. In fact their data that anti-ly6g reduces metastasis to barely detectable levels in both host and mutant genotypes is entirely consistent with this possibility. This will be independent of any form of adaptive immunity as it would occur in the first 24 hours after injection. To determine whether P2RX1 affects colony growth after seeding, Ly6G depletion at say 3-4 days after injection should be tested for its effect on colony growth. The differential may then be evident.

Response: We appreciate the reviewer's insightful comments. As the reviewer noted, accumulating evidence has revealed the important role of neutrophil in tumor metastatic cascade: including tumor extravasation, CTCs escort and capture, ECM remodeling, angiogenesis, and immunosuppression^{7, 11, 12, 13, 14, 15}. Neutrophil extracellular traps (NETs) formation is one of the most studied mechanisms, and have attracted much attention^{16, 17, 18, 19}.

To assess whether P2RX1 could influence the NETs formation, we performed immunofluorescent experiments. As previously reported, we observed that NETs were intensively generated in liver metastases. However, no significant differences were seen between WT and *P2rx1*^{-/-} mice (Figure S7C). Following the reviewer suggestion, we depleted neutrophils at 4 days after KPC cells injection. We found that only minimal reduction of liver metastases was observed in WT mice (Figure S5H). We think that the reviewer is right, neutrophils are involved in early biological events of metastatic cascades. Interestingly, though neutrophil depletion in WT mice at 4 days failed to achieve a similar effect, significant reduction of metastatic burden was

still obtained in *P2rx1*^{-/-} mice as compared to non-depletion mice. No differences of metastatic tumor growth were observed between WT and *P2rx1*^{-/-} mice receiving anti-Ly6G antibodies at 4 days. This result is similar to our previous neutrophil depletion result (Figure S5G), indicating that P2RX1-involved PDAC liver metastasis may be neutrophil-associated.

Fig 4 examines the numbers of Ly6G+ cells in the BM and the blood in naïve mice compared to mice bearing liver metastases with increased positive cells in the BM and in the blood. The level of cells 8.5% in the blood seems a bit low for mice.

Response: We thank the reviewer for this valuable comment. As shown in a public Mouse Phenome Project database (<https://phenome.jax.org/measures/31805>), different mouse strains may have significant variations in the percent of blood neutrophils. The neutrophil percent of C57BL/6 mice is approximately 10%. In the present study, C57BL/6 mice were used and our results is similar to the public data.

They treat isolated BM neutrophils with various cytokines and examine gene expression seeing a decrease in expression. This was only performed once so that statistical significance is not available.

Response: We thank the reviewer's helpful comment. To further confirm the results of RNA sequencing, RT-qPCR was performed. New results are added in the revised

manuscript (Figure S6E). The RT-qPCR results were similar with RNA sequencing results, indicating that P2RX1 was downregulated after various cytokines stimulation.

Because the neutrophils in WT mice that are P2RX1^{-/-} may be different from P2RX1^{-/-} neutrophils, some demonstration that the differences are consistent might be helpful while understanding that obtaining sufficient material might be difficult. IHC might help.

Response: We thank the reviewer for this valuable suggestion. As the reviewer notes, obtaining sufficient materials for comparing the differences is difficult. IHC is an efficient approach for semiquantitative analysis, yet not sensitive enough for comparing the differences between P2RX1⁻ neutrophils in WT mice and *P2rx1*^{-/-} neutrophils in P2RX1 KO mice. Therefore, for quantitative analysis, we performed FACS to isolate P2RX1⁺ and P2RX1⁻ neutrophils from liver metastatic WT mice, and *P2rx1*^{-/-} neutrophils from *P2rx1*^{-/-} mice. RT-qPCR was conducted to analysis the differences of TNF- α and PD-L1 expression. We observed that P2RX1⁺ neutrophils expressed the highest TNF- α , and the lowest PD-L1 (Figure S7B). The expression pattern of TNF- α and PD-L1 was similar between P2RX1⁻ and *P2rx1*^{-/-} neutrophils, which indicated the consistent tendency of genetic alterations of P2RX1⁻ neutrophils in WT mice and *P2rx1*^{-/-} neutrophils in *P2rx1*^{-/-} mice.

Fig. 5 neutrophils were isolated from liver metastases in WT mice and sorted to generate a population of P2RX1⁺ neutrophils to compare to P2RX1⁻ neutrophils isolated from liver metastases in P2RX1^{-/-} mice in gene expression finding enhanced glycolytic pathway genes in P2RX1⁺. They then examined ECAR and OCR in these neutrophil populations seeing enhanced OCR and decreased ECAR in neutrophils from the P2RX1^{-/-} liver metastases after stimulation with LPS or IL-4, results consistent with the gene expression differences. They also determine mitochondrial mass using mitotracker green. Because mitotracker green measures redox this is only a reflection of mitochondrial mass in the two populations at the same redox state. In this case where OCR differs so the redox states will differ, it would be more

appropriate to use a marker of mitochondrial mass, such as determination of a mitochondrial protein or amount of DNA etc to verify the conclusion about difference in mass.

Response: We thank the reviewer for this helpful comment. Following the reviewer's suggestion, we examined the mitochondrial DNA (mtDNA)/nuclear DNA (nDNA) ratio in WT and *P2rx1*^{-/-} neutrophils as previous reported methods²⁰. Our results showed that in both N1- and N2-polarized neutrophils, mtDNA/nDNA ratio was upregulated in *P2rx1*^{-/-} neutrophils compared to WT neutrophils (Figure S7H). These data were consistent with our previous MitoTracker Green studies (Figure S7G), suggesting an enhanced mitochondrial mass in *P2rx1*^{-/-} neutrophils.

In Fig 6 they now ask whether the difference in Nrf pathways found in the gene expression analysis might be directed by P2RX1 signaling. Fig. S7 showed that Nrf2 RNA was greater in P2RX1^{-/-} neutrophils than P2RX1⁺ tumor derived neutrophils and that stimulation with a variety of cytokines led to increased Nrf2 to a greater extent in P2RX1⁻ than P2RX1⁺ neutrophils. In fact a Nrf2 inhibitor reduced the response of increased OCR or decreased EACR to LPS+IFN γ or IL-4. IL-1 and TNF expression were affected.

They now ask about differences in PD-L1 expression on the different neutrophil populations. Fig 7 shows that a substantial proportion of CD8 cells from P2RX1^{-/-} mice in the liver metastases express PD-L1 more than from WT liver metastases. They do not show the expression on the KPC cells, which have been found to express PD-L1 in other settings.

Response: We thank the reviewer for this valuable comment.

By activating T cell-expressed PD-1, tumor and/or leukocyte-derived PD-L1 and PD-L2 induce immune exhaustion and facilitate tumor evasion. As the reviewer notes, in addition to neutrophils, KPC cells-derived PD-L1 may also contribute to the binding of CD8 T cell-expressed PD-1. Interestingly, we noticed that a well-executed study from Ingunn's team indicated that different tumor cell clones isolated from KPC mice may exhibit distinct phenotypes²¹. They observed that PD-L1 expression was

negligible in autochthonous KPC tumor and orthotopic tumor induced by a strain of isolated primary tumor epithelial cells (TECs) from KPC mice. However, in another KPC2a clone, PD-L1 expression is detected and α PD-L1 therapy is effective in KPC2a cell-induced orthotopic tumor. By revisiting the flow cytometry studies that examining PD-L1 expression in bulk liver metastatic tissues, we observed that neutrophils consisted most of the PD-L1+ cells (Figure S9B). In addition, we further performed immunofluorescence studies and found that PD-L1 was mainly distributed in Ly6G+ neutrophils (Figure S9C). These results indicate that neutrophil but not KPC cell is the main contributor of PD-L1.

Neutrophils from P2RX1^{-/-} liver metastases also expressed more PD-L1 than those from P2RX1^{+/+} mice. They then make the case that Nrf2 controls PD-L1 expression in neutrophils. Inhibition of Nrf2 blocks induction of PD-L1 by GM-CSF or tumor conditioned medium.

They show that making a prediction of the PD-L1 promoter based on integrative genomics that binding to the predicted regions can be found for Nrf2. Here one might ask for more demonstration that the presumed regions are actually part of the promoter activity before coming to the conclusion that Nrf2 directly regulated PD-L1 transcription.

Response: We thank the reviewer for raising this concern. To further confirm that Nrf2 can directly bind to PD-L1 promoter and regulate PD-L1 transcription, we performed luciferase reporter assay. Briefly, we cloned WT PD-L1 promoter into pGLB3 luciferase reporter vector. We found that co-transfection of Nrf2 markedly increased the promoter activity in NIH3T3 cells (Figure S10A). In contrast, mutation of PD-L1 at Peak1 did not response to Nrf2 co-transfection. These results suggest that PD-L1 is a direct transcriptional target of Nrf2. Detailed methods and results (Figure S10A) have now been provided in the revised manuscript.

They also use activated Tcells to show that addition of isolated neutrophils can reduce their proliferation with a reduction from 42% to 25% by the WT but 19% by the P2RX1^{-/-} neutrophils. This seems unlikely to be significant.

Response: We appreciate the reviewer's helpful comments. In the manuscript, results are presented as mean \pm standard deviation and shown with histograms. Since that visual bias may influence the result identification, figures in the manuscript have now been updated with dot pots to indicate each value, and raw data have been provided in the Source Data.

Anti PD-L1 also led to a reduction. However they do not indicate whether their T cells express PD-L1. A control with added anti PD-L1 is not shown. The same problem of potential T cell PD-L1 holds in the cytotoxicity assay as well as PD-L1 on the cancer cells.

Response: As the reviewer notes, recent studies have revealed that PD-L1 expressed in lymphocytes may also influence their cytotoxicity activity²². Following your suggestion, anti-PD-L1 neutralizing antibody has been added in lymphocyte proliferation and cytotoxicity studies. New results indicated that administration of anti-PD-L1 antibody had similar effects as anti-PD-1 antibody. The results have now been provided in the revised manuscript (Figure S10E-F).

Minor comments:

The sentence line 31 "Mechanistically, the transcription factor Nrf2 was upregulated in P2RX1-deficient neutrophils and accounted for PD-L1 expression and metabolic reprogramming." I feel is an overstatement. I think they have demonstrated a correlation but not causation.

Response: We thank the reviewer for this comment. We have revised this sentence in the revised manuscript as following:

"Mechanistically, the transcription factor Nrf2 was upregulated in P2RX1-deficient neutrophils and associated with PD-L1 expression and metabolic reprogramming."

Line 78 what is a “ complete” microenvironment?

Response: Here, we wanted to specify that the transcriptome data was not draw from microdissected tumor cells, but from bulk tumor tissues that contained tumor cells as well as the surrounding microenvironment tissues. We thank the reviewer for pointing this, and to avoid the misunderstanding, we have now revised this sentence in Results section and a similar sentence in the Induction section as following:

“To investigate specific transcriptome alterations that might influence PDAC liver metastases and its microenvironment, bioinformatics analyses were performed using a Gene Expression Omnibus (GEO) dataset (GSE71729) containing bulk-tissue microarray transcriptome data from primary PDAC, adjacent pancreas, liver metastatic PDAC and adjacent liver tissues (Figure S1A).”

“However, when comparing the transcriptome data of primary and metastatic tumors that contain microenvironment tissues, major differences were uncovered.”

Line 111 The references for the statement “The roles of ADORA2B in promoting PDAC progression 111 and tumor metastasis have been well characterized” but 14, 15 don’t really support this statement as regards pancreatic cancer.

Response: We appreciate the reviewer’s accuracy and apologize for the misuse of references. In the revised manuscript, this sentence has been revised as following:

“The roles of ADORA2B in promoting tumor metastasis have been well characterized”

Misspelling pancreas in SF1

Fig 6 Cannot really see blue filled vs black

Response: Thanks for your careful review. These mistakes haven been corrected in the revised manuscript.

Line 336 “These results suggest that increased intracellular ROS in P2rx1^{-/-} neutrophils contributes to PD-L1 expression by activating the ROS-sensitive transcription factor Nrf2.” This seems to overstate the correlation described

Line -362 “By upregulating the expression of PD-L1 in a ROS/Nrf2dependent manner, P2RX1- neutrophils exhibited potent immunosuppressive effects on PDAC- specific CTLs.” Again overstates the correlation shown.

Response: Thanks for your important comments. We have revised these first sentence as following:

“These results suggest that increased intracellular ROS in *P2rx1*^{-/-} neutrophils may favor the elevated PD-L1 expression by activating ROS-sensitive transcription factor Nrf2.”

The second sentence has been removed in the revised manuscript.

Could have been more in depth exploration of the connection between P2RX1 and NRF

Response: Thanks for your helpful comments. The connections between P2RX1 and NRF have now been thoroughly discussed in the Discussion section of the revised manuscript as following:

“By regulating transcriptional network of antioxidant, anti-inflammatory and metabolism, Nrf2 acts as a master transcription factor that controls adaptive responses to various environmental stresses³⁵. In a severe infection murine model, Nrf2 was found to restrain nuclear factor-κB-involved innate immune responses in myeloid macrophages and neutrophils⁵⁰. Our previous study observed that autocrine-induced activation of P2RX1 favored bacterial LPS-induced neutrophil activation¹³. However, the relation between Nrf2 and P2RX1 in neutrophils is unclear. In the present study, we showed that Nrf2 activity was upregulated in PDAC liver metastasis-infiltrated P2RX1- neutrophils. As a redox-sensitive transcription factor, we suggested that elevated ROS generation in P2RX1- neutrophil might contribute to the enhanced Nrf2 activity. Using chemical antagonist to block Nrf2, we observed that immunosuppressive phenotypes, including upregulated PD-L1 expression and

suppressed TNF- α expression were significantly restored in P2RX1- neutrophils. Therefore, the present study reveals that neutrophil Nrf2 activity can be elevated as a consequence of P2RX1 deficiency, and enhanced Nrf2 in P2RX1- neutrophils is essential for mediating suppression on antitumor immune responses in PDAC liver metastasis.”

Line 376 “Considering that glycolysis frequently supports immunostimulatory responses, whereas mitochondrial OXPHOS frequently supports immunosuppressive responses, we suggest that Nrf2 supports the plasticity of the immunosuppressive phenotype of P2RX1-378 deficient neutrophils.” I don’t understand what this means?

Response: Thanks for your comments. We apologize for the confusing expression. In this part, we intended to clarify that Nrf2-involved neutrophil metabolism may be associated with the shift of immune phenotype. We have revised this sentence as following:

“Considering that glycolysis frequently supports immunostimulatory responses, whereas mitochondrial OXPHOS frequently supports immunosuppressive responses, we suggest that Nrf2-supported mitochondrial metabolism in P2RX1-deficient neutrophils may be associated with the shaping of their immunosuppressive phenotypes.”

Line 389 “In the present study, we identified that liver metastatic PDAC evades antitumor immunity in the presence of accumulated extracellular ATP by mobilizing and recruiting P2RX1- neutrophils.” Don’t actually show this- it’s a hypothesis for the discussion.

Response: We apologize for the inappropriate expression. We have revised the inappropriate part in the revised manuscript as following:

“In the present study, we identified that liver metastatic PDAC evades antitumor immunity by mobilizing and recruiting P2RX1- neutrophils.”

Line 392 “As heterogeneous expression of purinergic receptors is also reported in other immune cell types, this strategy is likely common for other pathological conditions. “

What does this mean?

Response: Thanks for your comments. We apologize for the confusing expression. To better characterize our intended meaning, this sentence has been revised as following:

“Given that heterogeneous expression of purinergic receptors is also reported in other immune cell types²³, we speculate that the strategy to accumulate certain purinergic receptor-positive or -negative immune cell subtype may be adopted in other pathological conditions as well.”

Reviewer #3 (Remarks to the Author):

In this paper the authors described a role for P2RX1-negative neutrophils in pancreatic cancer liver metastases. Starting from bioinformatic analysis of available data base, they showed an association between P2RX1 expression and immunosuppressive microenvironment in PDAC liver metastasis. Authors confirmed these findings by performing in vivo experiments on P2RX1 knockout mice, in which they showed that the absence of P2RX1 led to an elevated level of PDAC liver metastasis. Flow cytometry analysis demonstrated that among all leukocyte infiltrating tumor microenvironment, the expression of P2RX1 is found higher in neutrophils. They continued with RNA seq analysis and characterized P2RX1-negative neutrophils. The authors suggested that the transcription factor Nrf2 is crucial in the immunosuppressive role of P2RX1-negative neutrophils by upregulating PD-L1 expression.

The manuscript is well presented, and the figures are organized well enough to catch the different points quickly.

Overall, the paper is interesting but some revisions are needed.

1: Considering P2RX1 as the key element in this manuscript, there is a need for a short description on purinergic receptors and their role in cancer or leukocyte trafficking in the introduction.

Response: We thank the reviewer for the helpful comment. A description of purinergic signaling in tumor microenvironment has now been provided as following: “Extracellular nucleotides (particularly ATP) and adenosine (ADO) are major biochemical constituents of tumor microenvironment⁹. By activating tumoral purinergic P2- and P1- receptors, extracellular purines facilitate the tumor growth directly¹⁰. In addition, host immune cells express abundant P2- and P1- receptors, and extracellular purines are essential for mediating immune cells trafficking and immune phenotypes¹¹.”

2: A recent paper (Patel, S et al. Nat Immunology 2019) showed that autocrine ATP signaling in neutrophils through purinergic receptors shape their activation state. Why this paper is not discussed?

Response: We appreciate the reviewer for bringing this highly relevant reference to our attention. This reference has been discussed in the Discussion section of revised manuscript as following:

“Purinergic signaling is crucial for regulating neutrophil immune functions⁴⁸. A recent study revealed that in the early-stage of cancer, bone marrow neutrophils ATP production was upregulated and autocrine ATP-involved purinergic signaling favored neutrophil spontaneous migration⁴⁹. However, in the late-stage of cancer, ATP production and migratory activity were indistinguishable, and the neutrophils displayed potent immunosuppressive activity⁴⁹. This evidence indicated that varied purinergic signaling may lead to heterogeneous neutrophil functions in tumor progression⁴⁹.”

3: Line 84-85: The authors claimed to analyze the immunome and quoted the reference 11 (Charoentong P. et al). Charoentong P. et al. announced 28 leukocytes while in this manuscript, 26 of them have been analyzed. More importantly in this analysis neutrophils are lacking, as neutrophils are the center of this manuscript they should be included in this analysis.

4: Even though later the authors described P2RX1 to be highly expressed in neutrophils, in their bioinformatic analysis in which they showed P2RX1 was positively correlated with antitumor immunocyte infiltration, they do not show the analyses for neutrophils (Figure S2A). Again, as neutrophils are important in the study, they should be inserted. Perhaps neutrophils are “hidden” with MDSC?

Response to comments 3 and 4: We appreciate the reviewer for noticing that this important information is missing. Neutrophils and eosinophils have now been included in Fig. 1B and Fig. S2A. Interestingly, as the reviewer pointed, we observed that neutrophil infiltration pattern was similar to that of MDSC. We have added comments on this results in the revised manuscript as following:

“Emerging evidence suggests that neutrophils play important roles in tumor metastasis¹⁵. We found that neutrophils were rare in primary PDAC and adjacent pancreas (Figure 1B). However, in PDAC liver metastases and adjacent liver tissues, neutrophils were among the most abundant immunocytes, which suggested that they might have potential roles in PDAC liver metastasis.”

5: In the Figure 5 or S6B, what is the relevance to stimulate neutrophils with IFN γ , IL4 and LPS for your model? In particular for LPS?. These in vitro experiments should be performed with tumor conditioned medium as performed in other parts of the paper.

Response: We thank the reviewer for raising this important question.

Based on the RNA-seq results of isolated neutrophils, we observed that metabolic and immune status were significantly varied between P2RX1+ and *P2rx1*^{-/-} neutrophils. Specifically, P2RX1+ neutrophils were characterized with N1-like markers, whereas *P2rx1*^{-/-} neutrophil were characterized with N2-like markers. We speculate that P2RX1 may be associated with neutrophil polarization, since differentially polarized neutrophils display distinct metabolic and immune phenotypes. Therefore, we used the stimulus (LPS, IFN- γ and IL-4) that commonly used to induce neutrophil polarization to verify whether P2RX1 could manipulate neutrophil polarization. As suggested, we found that metabolism and immune responses were altered in P2RX1-polarized neutrophils.

Here, we appreciate the reviewer’s valuable comment for strengthening the results by using TCM as stimuli. Following the reviewer’s suggestion, we have performed additional experiments and these results are shown in revised Figure 5. Briefly, we found that TCM significantly increased OCR in WT neutrophils, and which was further enhanced in *P2rx1*^{-/-} neutrophils (Figure 5C). In comparison, ECAR in WT neutrophils was mildly increased by TCM, and which was less obvious in *P2rx1*^{-/-} neutrophils (Figure 5C). In addition to metabolic shift, we found that TCM could induce higher PD-L1 expression in *P2rx1*^{-/-} neutrophils, and lower TNF- α expression in *P2rx1*^{-/-} neutrophils (Figure 5D).

6: Authors should insert references for the N1 and N2 markers they have chosen. Recent evidence has shown that MET signaling in neutrophils can induce also an immunosuppressive phenotype (Glodde, N. et al. *Immunity* 2017).

Response: We appreciate the reviewer's accuracy in this regard. We used MET as an antitumor N1 marker based on the previously published paper "MET is required for the recruitment of anti-tumoural neutrophils. *Nature*. 2015;522(7556):349-353". References of the N1 and N2 markers that we choose have been provided in the revised manuscript.

7: The phenotype of CXCR4+ P2RX1-negative neutrophils could be explored more deeply. Are they immature neutrophils, aged neutrophils, or mature activated neutrophils?

Response: We thank the reviewer for this insightful comment. Neutrophils are originally recognized as homogeneous cells that participate in anti-infection immunity. In recent years, accumulating evidence has revealed the abundant heterogeneities of neutrophils²⁴. As the reviewer mentioned, several classifications including "mature or immature", and "fresh or aged" have been adopted to characterize the neutrophil heterogeneity. Based on previous reports, CXCR2 is used to identify "mature" or "immature" neutrophils²⁵, and CXCR4 is used to identify "fresh" or "aged" neutrophils²⁶. It is noticed that, neutrophils ageing occurs when they are released from bone marrow, and CXCR4 is commonly used to evaluate neutrophil ageing in peripheral tissues but not in bone marrow^{27, 28}. In our original study, CXCR4 expression was analyzed in bone marrow neutrophils for investigating neutrophil homeostasis. Therefore, as the reviewer suggests, to further study the relation between P2RX1 expression and the known neutrophil heterogeneity markers, we have performed additional experiments in liver metastatic neutrophils.

In liver metastatic tissues, we found almost all neutrophils were CXCR2+ (Figure S6A), which suggested that both P2RX1+ and P2RX1- neutrophils were mature neutrophils. Interestingly, we noticed CXCR2 expression intensity was different

between P2RX1+ and P2RX1- neutrophils, with P2RX1- neutrophil expressing higher level of CXCR2 (Figure S6A-B). Given that CXCR2 is an important chemokine receptor for regulation neutrophil migration, we suppose that higher CXCR2 may favor the enrichment of P2RX1- neutrophils in liver metastases. In addition, we observed that CXCR4 expression was different between P2RX1+ and P2RX1- neutrophils, with P2RX1- neutrophils expressing less CXCR4 (Figure S6C-D). It suggested that P2RX1- neutrophils were “fresher” than P2RX1+ neutrophils. Therefore, our results indicate that P2RX1- neutrophils in PDAC liver metastases are characterized with mature and fresh phenotypes. These results have been added in the Results section of the revised manuscript.

Minor comments

1. Along the text whenever some results are described, the corresponding figure, or a reference, should be mentioned, here are some examples in which in this manuscript this aspect is ignored:

- Page5, line 90

- Page7, line 151

- Page10, line 201 (Figure 4 is not describing the mentioned information)

Response: We apologize for these omissions. We have combed through the manuscript to make corrections.

2. In figure 3A the results are shown only by histograms, but as in the text the frequency of cells expressing P2RX1 is also mentioned, it would be useful to indicate the percentage of positive cell in the histograms.

Response: Thanks for pointing this issue. Percentage of P2RX1 positive cell has now been added in Figure S5B of the revised manuscript.

References:

1. Ledderose C, *et al.* Cutting off the power: inhibition of leukemia cell growth by pausing basal ATP release and P2X receptor signaling? *Purinergic signalling* **12**, 439-451 (2016).
2. Ledderose C, *et al.* Mitochondrial Dysfunction, Depleted Purinergic Signaling, and Defective T Cell Vigilance and Immune Defense. *The Journal of infectious diseases* **213**, 456-464 (2016).
3. Wang X, *et al.* Endotoxin-induced autocrine ATP signaling inhibits neutrophil chemotaxis through enhancing myosin light chain phosphorylation. *Proceedings of the National Academy of Sciences of the United States of America* **114**, 4483-4488 (2017).
4. Mallilankaraman K, *et al.* MCUR1 is an essential component of mitochondrial Ca²⁺ uptake that regulates cellular metabolism. *Nature cell biology* **14**, 1336-1343 (2012).
5. Santulli G, Xie W, Reiken SR, Marks AR. Mitochondrial calcium overload is a key determinant in heart failure. *Proceedings of the National Academy of Sciences of the United States of America* **112**, 11389-11394 (2015).

6. Huang KF, Ma KH, Jhap TY, Liu PS, Chueh SH. Ultraviolet B irradiation induced Nrf2 degradation occurs via activation of TRPV1 channels in human dermal fibroblasts. *Free radical biology & medicine* **141**, 220-232 (2019).
7. Mollinedo F. Neutrophil Degranulation, Plasticity, and Cancer Metastasis. *Trends in immunology* **40**, 228-242 (2019).
8. Mulryan K, *et al.* Reduced vas deferens contraction and male infertility in mice lacking P2X1 receptors. *Nature* **403**, 86-89 (2000).
9. Kimura Y, *et al.* The innate immune receptor Dectin-2 mediates the phagocytosis of cancer cells by Kupffer cells for the suppression of liver metastasis. *Proceedings of the National Academy of Sciences of the United States of America* **113**, 14097-14102 (2016).
10. Yu X, *et al.* Immune modulation of liver sinusoidal endothelial cells by melittin nanoparticles suppresses liver metastasis. *Nature communications* **10**, 574 (2019).
11. Szczerba BM, *et al.* Neutrophils escort circulating tumour cells to enable

- cell cycle progression. *Nature* **566**, 553-557 (2019).
12. Spiegel A, *et al.* Neutrophils Suppress Intraluminal NK Cell-Mediated Tumor Cell Clearance and Enhance Extravasation of Disseminated Carcinoma Cells. *Cancer discovery* **6**, 630-649 (2016).
 13. Wculek SK, Malanchi I. Neutrophils support lung colonization of metastasis-initiating breast cancer cells. *Nature* **528**, 413-417 (2015).
 14. Coffelt SB, *et al.* IL-17-producing gammadelta T cells and neutrophils conspire to promote breast cancer metastasis. *Nature* **522**, 345-348 (2015).
 15. Cools-Lartigue J, *et al.* Neutrophil extracellular traps sequester circulating tumor cells and promote metastasis. *The Journal of clinical investigation*, (2013).
 16. Lee W, Ko SY, Mohamed MS, Kenny HA, Lengyel E, Naora H. Neutrophils facilitate ovarian cancer premetastatic niche formation in the omentum. *The Journal of experimental medicine* **216**, 176-194 (2019).

17. Inoue M, *et al.* Plasma redox imbalance caused by albumin oxidation promotes lung-predominant NETosis and pulmonary cancer metastasis. *Nature communications* **9**, 5116 (2018).
18. Park J, *et al.* Cancer cells induce metastasis-supporting neutrophil extracellular DNA traps. *Science translational medicine* **8**, 361ra138 (2016).
19. Yang L, *et al.* DNA of neutrophil extracellular traps promotes cancer metastasis via CCDC25. *Nature* **583**, 133-138 (2020).
20. Zhang Y, *et al.* Listeria hijacks host mitophagy through a novel mitophagy receptor to evade killing. *Nature immunology* **20**, 433-446 (2019).
21. Burrack AL, Spartz EJ, Raynor JF, Wang I, Olson M, Stromnes IM. Combination PD-1 and PD-L1 Blockade Promotes Durable Neoantigen-Specific T Cell-Mediated Immunity in Pancreatic Ductal Adenocarcinoma. *Cell reports* **28**, 2140-2155 e2146 (2019).
22. Diskin B, *et al.* PD-L1 engagement on T cells promotes self-tolerance and suppression of neighboring macrophages and effector T cells in

- cancer. *Nature immunology* **21**, 442-454 (2020).
23. Bjorklund AK, *et al.* The heterogeneity of human CD127(+) innate lymphoid cells revealed by single-cell RNA sequencing. *Nature immunology* **17**, 451-460 (2016).
 24. Ng LG, Ostuni R, Hidalgo A. Heterogeneity of neutrophils. *Nature reviews Immunology* **19**, 255-265 (2019).
 25. Evrard M, *et al.* Developmental Analysis of Bone Marrow Neutrophils Reveals Populations Specialized in Expansion, Trafficking, and Effector Functions. *Immunity* **48**, 364-379 e368 (2018).
 26. Casanova-Acebes M, *et al.* Rhythmic modulation of the hematopoietic niche through neutrophil clearance. *Cell* **153**, 1025-1035 (2013).
 27. Zhang D, *et al.* Neutrophil ageing is regulated by the microbiome. *Nature* **525**, 528-532 (2015).
 28. Adrover JM, *et al.* A Neutrophil Timer Coordinates Immune Defense and Vascular Protection. *Immunity* **50**, 390-402 e310 (2019).

REVIEWERS' COMMENTS

Reviewer #1 (Remarks to the Author):

I found the Authors' reply fully satisfactory.

Reviewer #2 (Remarks to the Author):

The authors have addressed the previous comments to my satisfaction.

Reviewer #3 (Remarks to the Author):

The authors have addressed my concerns.

Reviewer #1 (Remarks to the Author):

I found the Authors' reply fully satisfactory.

Response: We appreciate the reviewer's positive comment.

Reviewer #2 (Remarks to the Author):

The authors have addressed the previous comments to my satisfaction.

Response: We thank the reviewer for the positive comment.

Reviewer #3 (Remarks to the Author):

The authors have addressed my concerns.

Response: We appreciate the reviewer for the positive comment.